# Adversarial Examples in Multi-Layer Random ReLU Networks

**Peter L. Bartlett**
Department of Electrical Engineering and Computer Science
Department of Statistics
UC Berkeley

**Sébastien Bubeck**
Microsoft Research Redmond

**Yeshwanth Cherapanamjeri**
Department of Electrical Engineering and Computer Science
UC Berkeley

## Abstract

We consider the phenomenon of adversarial examples in ReLU networks with independent Gaussian parameters. For networks of constant depth and with a large range of widths (for instance, it suffices if the width of each layer is polynomial in that of any other layer), small perturbations of input vectors lead to large changes of outputs. This generalizes results of Daniely and Schacham (2020) for networks of rapidly decreasing width and of Bubeck et al (2021) for two-layer networks. Our proof shows that adversarial examples arise in these networks because the functions they compute are *locally* very similar to random linear functions. Bottleneck layers play a key role: the minimal width up to some point in the network determines scales and sensitivities of mappings computed up to that point. The main result is for networks with constant depth, but we also show that some constraint on depth is necessary for a result of this kind, because there are suitably deep networks that, with constant probability, compute a function that is close to constant.

## 1   Introduction and Main Result

Since the phenomenon of adversarial examples was first observed in deep networks [SZS+14], there has been considerable interest in why this extreme sensitivity to small input perturbations arises in deep networks [GSS15, SSRD19, BLPR19, DS20, BCGdC21] and how it can be detected and avoided [CW17a, CW17b, FCSG17, MMS+18, QMG+19]. Building on the work of Shamir et al [SSRD19], Daniely and Schacham [DS20] prove that small perturbations (measured in the Euclidean norm) can be found for any fixed input and most Gaussian parameters in certain ReLU networks—those in which each layer has vanishing width relative to the previous layer—and conjectured the same result without this strong constraint on the architecture. Bubeck, Cherapanamjeri, Gidel and Tachet des Combes [BCGdC21] prove that the same phenomenon occurs in general two-layer ReLU networks, and give experimental evidence of its presence in deeper ReLU networks.

In this paper, we prove that adversarial examples also arise in deep ReLU networks with random weights for a wide variety of network architectures—those with constant depth and polynomially-related widths. The key fact underlying this phenomenon was already observed in [DS20]: a high-dimensional linear function $f(x) = w^\top x$ with input $x \neq 0$ and random parameter vector $w$ with a uniformly chosen direction will satisfy $\|\nabla f(x)\| \|x\| \gg |f(x)|$ with high probability. This implies

the existence of a nearby adversarial example for this linear function: a perturbation of $x$ of size $|f(x)|/\|\nabla f(x)\| \ll \|x\|$ in the direction $-f(x)\nabla f(x)$ will flip the sign of $f(x)$. This observation can be extended to nonlinear functions that are locally almost linear. Indeed, it is easy to show that for all $x, u \in \mathbb{R}^d$,

$$|f(x+u) - (f(x) + \langle u, \nabla f(x)\rangle)| \leq \|u\| \sup\left\{\|\nabla f(x) - \nabla f(x+v)\| : v \in \mathbb{R}^d, \|v\| \leq \|u\|\right\},$$

and thus to demonstrate the existence of an adversarial example near $x$ for a function $f$, it suffices to show the smoothness property:

$$\text{for all } v \in \mathbb{R}^d \text{ with } \|v\| \lesssim |f(x)|/\|\nabla f(x)\|, \qquad \|\nabla f(x) - \nabla f(x+v)\| \ll \|\nabla f(x)\|. \qquad (1)$$

We show that for a deep ReLU network with random parameters and a high-dimensional input vector $x$, there is a relatively large ball around $x$ where the function computed by the network is very likely to satisfy this smoothness property. Thus, adversarial examples arise in deep ReLU networks with random weights because the functions that they compute are very close to linear in this sense.

It is important to notice that ReLU networks are not smooth in a classical sense, because the nondifferentiability of the ReLU nonlinearity implies that the gradient can change abruptly. But for the smoothness condition (1), it suffices to have $\|\nabla f(x) - \nabla f(x+v)\| \leq \epsilon + \phi(\|v\|)$, for some increasing function $\phi : \mathbb{R}_+ \to \mathbb{R}_+$, provided that $\epsilon + \phi(\|v\|) \ll \|\nabla f(x)\|$. We prove an inequality like this for ReLU networks, where the $\epsilon$ term decreases with width.

Consider a network with input dimension $d$, $\ell+1$ layers, a single real output, and complete connections between layers. Let $d_1, \ldots, d_\ell$ denote the dimensions of the layers. The network has independent random weight matrices $W_i \in \mathbb{R}^{d_i \times d_{i-1}}$ for $i \in [\ell+1]$, where we set $d_0 = d$ and $d_{\ell+1} = 1$. For input $x \in \mathbb{R}^d$, the network output is defined as follows:

$$f(x) = W_{\ell+1} \cdot \sigma(W_\ell \cdot \sigma(W_{\ell-1} \cdot \sigma(\cdots \sigma(W_1 \cdot x) \cdots))) \text{ where } \sigma(x)_i = \max\{x_i, 0\}$$

$$W_{\ell+1} \sim \mathcal{N}(0, I/d_\ell) \text{ and } \forall i \in [\ell], (W_i)_{j,k} \overset{i.i.d}{\sim} \mathcal{N}(0, 1/d_{i-1}). \qquad \text{(NN-DEF)}$$

Note that the scale of the parameters is chosen so that all the real-valued signals that appear throughout the network have roughly the same scale. This is only for convenience: because the ReLU is positively homogeneous (that is, for $\alpha > 0$, $\sigma(\alpha x) = \alpha\sigma(x)$), the scaling is not important for our results; the $1/d_{i-1}$ in (NN-DEF) could be replaced by any constant without affecting the ratio between the norm of an input vector and that of a perturbation required to change the sign of the corresponding output.

The following theorem is the main result of the paper.

**Theorem 1.1.** *Fix $\ell \in \mathbb{N}$. There are constants $c_1, c_2, c_3$ that depend on $\ell$ for which the following holds. Fix $\delta \in (0,1)$ and let $f(\cdot)$ be an $(\ell+1)$-layer ReLU neural network defined by (NN-DEF) with input dimension $d$ and intermediate layers of width $\{d_i\}_{i=1}^\ell$. Suppose that the widths satisfy*

$$d_{\min} \geq c_1 (\log d_{\max})^{c_2} \log 1/\delta \text{ where } d_{\min} = \min\left\{\{d_i\}_{i=1}^\ell, d\right\}, d_{\max} = \max\left\{\{d_i\}_{i=1}^\ell, d\right\}.$$

*Then for any fixed input $x \neq 0$, with probability at least $1 - \delta$,*

$$|f(x + \eta\nabla f(x))| \geq |f(x)| \text{ and } \text{sign}(f(x + \eta\nabla f(x))) \neq f(x),$$

*for an $\eta$ satisfying*

$$\frac{\|\eta\nabla f(x)\|}{\|x\|} \leq c_3 \sqrt{\frac{\log 1/\delta}{d}}.$$

*It suffices to choose $c_1 = (C_1\ell)^{c_2}$, $c_2 = C_2\ell$, $c_3 = C_3^\ell$, for some absolute constants $C_1$, $C_2$, $C_3$.*

This theorem concerns networks of fixed depth, and the constants in the size of the perturbation and in the requirement on the network width are larger for deeper networks. We also prove a converse result that illustrates the need for some constraint on the depth. Theorem 3.1 shows that when the depth is allowed to grow polynomially in the input dimension $d$, the function computed by a random ReLU network is essentially constant, which rules out the possibility of adversarial examples.

The heart of the proof of Theorem 1.1 is to show a smoothness property like (1). It exploits a decomposition of the change of gradient between two input vectors. Define $H_i : \mathbb{R}^d \to \mathbb{R}^{d_i \times d_i}$

as $H_i(x)_{jk} = \mathbf{1}\{j = k, v_i(x)_j \geq 0\}$ with $v_i(x) = W_i\sigma(\cdots\sigma(W_1x))$. For two input vectors $x, y \in \mathbb{R}^d$, we will see in Section 2.4 that we can decompose the change of gradient as

$$\nabla f(x) - \nabla f(y) = \sum_{j=1}^{\ell} W_{\ell+1}\left(\prod_{i=\ell}^{j+1} H_i(x)W_i\right) \cdot (H_j(x) - H_j(y))W_j \cdot \left(\prod_{i=j-1}^{1} H_i(y)W_i\right).$$

(Here and elsewhere, indices of products of matrices run backwards, so $\prod_{i=j}^{k} M_i = I$ when $j < k$.) For the $j$th term in the decomposition, we need to control the scale of: the gradient of the mapping from the input to the output of layer $j$, the change in the layer $j$ nonlinearity $H_j(x) - H_j(y)$, and the gradient from layer $j$ to the output. It turns out that controlling these quantities depends crucially on the width of the narrowest layer before layer $j$—we call this the *bottleneck layer* for layer $j$. This width determines the dimension of the image at layer $j$ of a ball in the input space. In proving bounds on gradients and function values that hold uniformly over pairs of nearby vectors $x$ and $y$, this dimension—the width of the bottleneck layer—dictates the size of a discretization (an $\epsilon$-net) that is a crucial ingredient in the proof of these uniform properties. Our analysis involves working separately with the segments between these bottleneck layers. We show that for an input $x \in \mathbb{R}^d$ satisfying $\|x\| = \sqrt{d}$ and any $y$ in a ball around $x$, with high probability $\|\nabla f(x) - \nabla f(y)\| = o(1)$, but $|f(x)|$ is no more than a constant and $\|\nabla f(x)\|$ is at least a constant. This implies the existence of a small ($o(\|x\|)$) perturbation of $x$ in the direction $-f(x)\nabla f(x)$ that flips the sign of $f(x)$.

These results suggest several interesting directions for future work. First, our results show that for high-dimensional inputs, adversarial examples are inevitable in random ReLU networks with constant depth, and unlikely in networks with polynomial depth. Beyond this, we do not know how the sensitivity to input perturbations decreases with depth. Similarly, both results are restricted to networks with subexponential width, and it is not clear what happens for very wide networks. Finally, we show that networks with random weights suffer from adversarial examples because their behavior is very similar to that of random linear functions. It would be worthwhile to determine whether randomly initialized trained networks retain this nearly linear behavior, and hence suffer from adversarial examples for the same reason.

**Related Work:** Related theoretical work include the recent result of Bubeck and Sellke [BS21] who, following up on Bubeck, Li and Nagaraj [BLN21], show that only *mildly* over-parameterized networks when trained on random data have large Lipschitz constants. While these results apply to a broader class of networks including those potentially trained on data, this weaker property on its own does not suffice to explain the prevalence of adversarial examples and computational ease of finding them. Indeed, establishing this fact requires understanding the behavior of the local landscape of the function computed by the network which these approaches do not capture.

Closely related empirical works include work by Madry, Makelov, Schmidt, Tsipras and Vladu [MMS+18] and by Qin, Martens, Gowal, Krishnan, Dvijotham, Fawzi, De, Stanforth and Kohli [QMG+19]. We note that both these works emphasize the importance of the interplay between the local behavior of the function and the existence of adversarial perturbations. [MMS+18] identify local near-linearity as a cause of adversarial examples and propose a robust training procedure attempting to eliminate this property. On the other hand, [QMG+19] suggest an alternative robust training procedure that retains local near-linearity but eliminates adversarial examples by ensuring that the learnt network (despite being locally linear) is robust to single-step perturbations. Our work lends theoretical grounding to this phenomenon showing that local linearity arises naturally at initialization.

## 2 Proof of Main Theorem

In this section, we provide an outline of the proof of Theorem 1.1. As described above, we will prove our result first by showing that the gradient at $x$ has large norm and changes negligibly in a large ball around $x$. The first condition is established in Subsection 2.1. The second step is more intricate. First, we prove the decomposition of the gradient differences in Subsection 2.2. Then, in Subsection 2.3, we track the scale of the ball around $x$ as it propagates through the network. Finally, in Subsection 2.4,

we use this result to bound the terms in the decomposition of the gradient differences to show that our network is locally linear. For the rest of the proof, unless otherwise stated, we consider a fixed $x \in \mathbb{R}^d$, and we assume:

$$d_{\min} \geq (C\ell \log d_{\max})^{240\ell} \log 1/\delta, \quad \|x\| = \sqrt{d} \text{ and } R \coloneqq \frac{\sqrt{d_{\min}}}{(\ell \log d_{\max})^{80\ell}} = \Omega\left((\ell \log d_{\max})^{40\ell}\right).$$

Additionally, we randomize the activations of neurons whenever they receive an input of 0. This does not change the behavior of the neural network in terms of its output or the images of the input through the layers of the network but greatly simplifies our proof. For $x \in \mathbb{R}^d$, we let:

$$x_0 \coloneqq x, \ \widetilde{f}_i(x) \coloneqq W_i f_{i-1}(x), \ (D_i(y))_{j,k} = \begin{cases} 1 & \text{w.p } \frac{1}{2} \text{ if } j = k, \ y_j = 0, \\ 1 & \text{if } j = k, \ y_j > 0, \\ 0 & \text{otherwise,} \end{cases} \quad f_i(x) = D_i(\widetilde{f}_i(x))\widetilde{f}_i(x).$$

Our first key observation is that the randomization in the activation units allows us the following distributional equivalences, proved in [Appendix A.1](#).

**Lemma 2.1.** *Let $m \in \mathbb{N}$, $\{d_i\}_{i=0}^m \subset \mathbb{R}^d$ and $W_i \in \mathbb{R}^{d_i \times d_{i-1}}$ be distributed such that each entry of $W_i$ is drawn iid from any symmetric distribution. Then, defining for $x \in \mathbb{R}^{d_0}$:*

$$\begin{aligned} h_0(x) &= x, \\ \widetilde{h}_i(x) &= W_i h_{i-1}(x) \qquad \text{where } (D_i(y))_{i,j} = \begin{cases} 1, & \text{if } j = k \text{ and } y_j > 0 \\ 1, & \text{with probability } \frac{1}{2} \text{ if } j = k, \ y_j = 0 \\ 0, & \text{otherwise} \end{cases} \\ h_i(x) &= D_i(\widetilde{h}_i(x))\widetilde{h}_i(x) \end{aligned}$$

*we have the distributional equivalences for any $x \neq 0$ and fixed diagonal matrices $B_1, \ldots, B_m$:*

$$W_m \prod_{j=m-1}^{1} (D_j(\widetilde{h}_j(x)) + B_j)W_j \stackrel{d}{=} W_m \prod_{j=m-1}^{1} (D_j + B_j)W_j$$

$$\left\| \prod_{j=m}^{1} (D_j(\widetilde{h}_j(x)) + B_j)W_j \right\| \stackrel{d}{=} \left\| \prod_{j=m}^{1} (D_j + B_j)W_j \right\|$$

$$\text{where } (D_j)_{k,l} = \begin{cases} 1, & \text{with probability } 1/2 \text{ if } k = l \\ 0, & \text{otherwise} \end{cases}$$

## 2.1 Concentration of Function Value and Gradient at a Fixed Point

We first present a simple lemma that shows that the gradient at $x$ is at least a constant and that its output value is bounded. The proof gives an illustration of how [Lemma 2.1](#) will be used through the more involved proofs in the paper.

**Lemma 2.2.** *For some universal constant c, with probability at least $1 - \delta$ we have:*

$$|f(x)| \leq c2^\ell \sqrt{\log 1/\delta} \text{ and } \|\nabla f(x)\| \geq \frac{1}{2^{\ell+1}}.$$

*Proof.* Note that

$$\nabla f(x) = W_{\ell+1} \prod_{i=\ell}^{1} D_i(\widetilde{f}_i(x))W_i \text{ and } f(x) = \nabla f(x)x.$$

And we have from [Lemma 2.1](#), $\nabla f(x) \stackrel{d}{=} \tilde{W}_{\ell+1} \prod_{i=\ell}^{1} D_i \tilde{W}_i$, where

$$(D_i)_{j,k} = \begin{cases} 1 & \text{with probability } 1/2 \text{ if } j = k, \\ 0 & \text{otherwise,} \end{cases} \qquad \{W_i\}_{i=1}^{\ell+1} \stackrel{d}{=} \{\tilde{W}_i\}_{i=1}^{\ell+1}.$$

Therefore, it suffices to analyze the random vector $\tilde{W}_{\ell+1} \prod_{i=\ell}^{1} D_i \tilde{W}_i$. We first condition on a favorable event for the $D_i$. Note that we have by the union bound and an application of Hoeffding's inequality (e.g., [BLM13, Theorem 2.8]) that:

$$\forall i \in [\ell] : \operatorname{Tr} D_i \geq \frac{d_i}{3} \text{ with probability at least } 1 - \delta/4,$$

since $d_i \geq c \log(4\ell/\delta)$. We now condition on the $D_i$ and note that:

$$\left\| \widetilde{W}_{\ell+1} \prod_{i=\ell}^{1} D_i \widetilde{W}_i \right\| \overset{d}{=} \left\| W_{\ell+1}^{\dagger} \prod_{i=\ell}^{1} W_i^{\dagger} \right\| \text{ where}$$

$$\forall i \in \{2, \ldots, \ell\} : W_i^{\dagger} \in \mathbb{R}^{\operatorname{Tr} D_i \times \operatorname{Tr} D_{i-1}}, \ W_{\ell+1}^{\dagger} \in \mathbb{R}^{1 \times \operatorname{Tr} D_\ell}, \ W_1^{\dagger} \in \mathbb{R}^{\operatorname{Tr} D_1 \times d}$$

$$(W_1^{\dagger})_{j,:} \sim \mathcal{N}(0, I/d), \ W_{\ell+1}^{\dagger} \sim \mathcal{N}(0, I/d_\ell), \ \forall i \in \{2, \ldots, \ell+1\} : (W_i^{\dagger})_{j,:} \sim \mathcal{N}\left(0, I/d_{i-1}\right).$$

From the above display, we obtain from [Theorem C.3](#) and its corollary [C.4](#),

$$\left\| W_{\ell+1}^{\dagger} \prod_{i=\ell}^{1} W_i^{\dagger} \right\| \geq \frac{1}{2} \cdot \left\| W_{\ell+1}^{\dagger} \prod_{i=\ell}^{2} W_i^{\dagger} \right\| \qquad \text{with probability at least } 1 - \delta/(4\ell)$$

$$\geq \frac{1}{2^{\ell+1}} \qquad \text{with probability at least } 1 - \delta/4 \text{ by induction,}$$

since $\min_{0 \leq i \leq \ell} d_i \geq c \log(4\ell/\delta)$. Through a similar argument, we obtain:

$$\left\| \tilde{W}_{\ell+1} \prod_{i=\ell}^{1} D_i \tilde{W}_i \right\| \leq 2^{\ell+1} \tag{2}$$

with probability at least $1 - \delta/4$. We also have:

$$\left( \tilde{W}_{\ell+1} \prod_{i=\ell}^{1} D_i \tilde{W}_i \ \Bigg| \ \left\| \tilde{W}_{\ell+1} \prod_{i=\ell}^{1} D_i \tilde{W}_i \right\| = m \right) \overset{d}{=} \operatorname{Unif}\left(m \mathbb{S}^{d-1}\right).$$

Combining this, [Eq. (2)](#) and [Lemma C.6](#), we get that:

$$|f(x)| \leq c 2^{\ell} \sqrt{\log 1/\delta}$$

with probability at least $1 - \delta/2$, for some absolute constant $c$. A union bound over all the preceeding events concludes the lemma. $\qquad \square$

## 2.2 A Decomposition of Local Gradient Changes

The following decomposition allows us to reason about deviations layer by layer:

$$\nabla f(x) - \nabla f(y) = W_{\ell+1} \left( \prod_{i=\ell}^{1} D_i(\widetilde{f}_i(x)) W_i - \prod_{i=\ell}^{1} D_i(\widetilde{f}_i(y)) W_i \right)$$

$$= W_{\ell+1} \left( \prod_{i=\ell}^{1} D_i(\widetilde{f}_i(x)) W_i - \left( D_\ell(\widetilde{f}_\ell(x)) + (D_\ell(\widetilde{f}_\ell(y)) - D_\ell(\widetilde{f}_\ell(x))) \right) W_\ell \prod_{i=\ell-1}^{1} D_i(\widetilde{f}_i(y)) W_i \right)$$

$$= W_{\ell+1} \left( D_\ell(\widetilde{f}_\ell(x)) W_\ell \left( \prod_{i=\ell-1}^{1} D_i(\widetilde{f}_i(x)) W_i - \prod_{i=\ell-1}^{1} D_i(\widetilde{f}_i(y)) W_i \right) \right.$$

$$\left. + (D_\ell(\widetilde{f}_\ell(x)) - D_\ell(\widetilde{f}_\ell(y))) W_\ell \prod_{i=\ell-1}^{1} D_i(\widetilde{f}_i(y)) W_i \right)$$

$$= \sum_{j=1}^{\ell} \underbrace{W_{\ell+1} \left( \prod_{i=\ell}^{j+1} D_i(\widetilde{f}_i(x)) W_i \right) \cdot (D_j(\widetilde{f}_j(x)) - D_j(\widetilde{f}_j(y))) W_j \cdot \left( \prod_{i=j-1}^{1} D_i(\widetilde{f}_i(y)) W_i \right)}_{\Delta_j}.$$

$$\text{(GD-DECOMP)}$$

We use this decomposition to show that the gradient is locally constant. Concretely, consider a fixed term in the above decomposition and let $i^* = \arg \min_{i < j} d_i$. Letting $M_{i^*} = \prod_{i=i^*}^{1} D_i(\widetilde{f}_i(y)) W_i$, we can bound a single term, $\Delta_j$, as follows:

$$\|\Delta_j\| \leq \left\| W_{\ell+1} \left( \prod_{i=\ell}^{j+1} D_i(\widetilde{f}_i(x)) W_i \right) (D_j(\widetilde{f}_j(x)) - D_j(\widetilde{f}_j(y))) W_j \left( \prod_{i=j-1}^{i^*+1} D_i(\widetilde{f}_i(y)) W_i \right) \right\| \|M_{i^*}\|.$$

We bound the above by bounding both of the two factors on the right-hand-side. In the above expression, the length of $\Delta_j$ is bounded by a product of the length of the corresponding factor in a truncated network starting at the output of layer $i^*$ and a product of masked weight matrices up to layer $i^*$ corresponding to the activation patterns of $y$. Intuitively, the first factor is expected to be small if the images of $x$ and $y$ at layer $i^*$ are close and hence, we show that an image of a suitably small ball around $x$ remains close to the image of $x$ through all the layers of the network (Lemma 2.3). We then prove a spectral norm bound on $M_{i^*}$ in Lemma 2.6 and finally, establish a bound on $\|\Delta_j\|$ in Lemma 2.8 where we crucially rely on the scale preservation guarantees provided by Lemma 2.3. Finally, combining these results with those of Subsection 2.1 completes the proof of Theorem 1.1.

## 2.3  Scale Preservation of Local Neighborhoods

In this section, we show that the image of a ball around $x$ remains in a ball of suitable radius around the image of $x$ projected through the various layers of the network. Here, we introduce additional notation used in the rest of the proof:

$$\forall j \in \{0\} \cup [\ell+1] : f_{j,j}(x) = x, \ \forall i > j : \widetilde{f}_{i,j}(x) = W_i f_{i-1,j}(x), \ f_{i,j}(x) = D_i(\widetilde{f}_{i,j}(x))\widetilde{f}_{i,j}(x).$$

We now describe the decomposition of the neural network into segments, which are bounded by what we call bottleneck layers, and our analysis works separately with these segments. This decomposition is crucial for reducing the sizes of the $\epsilon$-nets that arise in our proofs. Intuitively, when we construct an $\epsilon$-net to prove that some property holds uniformly over a ball, it is crucial to work with the lowest-dimensional image of that ball, which appears in a bottleneck layer. These bottleneck layers are denoted by indices $\{i_j\}_{j=1}^{m}$, defined recursively from the output layer backwards with the convention that $d_0 = d$:

$$i_1 := \arg\min_{i \leq \ell} d_i, \quad \forall j > 1 \text{ s.t } i_j \geq 1, i_{j+1} := \arg\min_{j < i_j} d_j, \quad d_{\min}^i = \min_{j < i} d_j. \quad \text{(NN-DECOMP)}$$

Note, that $i_m = 0$, for all $j \in [m-1]$, $d_{i_j} < d_{i_{j+1}}$ and for all $k \in \{i_{j+1}, \ldots, i_j - 1\}$, $d_k \geq d_{i_{j+1}}$.

The following technical lemma bounds the scaling of the images at a layer in the network of an input $x$ and of a ball around $x$. The crucial properties we will exploit are that these images avoid the origin, and that the radius of the image of the ball is not too large. The full proof is deferred to Appendix A.2. In the proof sketch, we carry out the section of the proof where we transition between bottleneck layers in full as it is a simple illustration of how such ideas are used through the rest of the proof.

**Lemma 2.3.** *We have with probability at least $1 - \delta$,*

$$\forall i \in [\ell] : \|f_i(x)\| \geq \frac{1}{2^i} \cdot \sqrt{d_i},$$

$$\forall \|x' - x\| \leq R : \|\widetilde{f}_i(x) - \widetilde{f}_i(x')\| \leq (C \log d_{\max})^{i/2} \cdot \left(1 + \sqrt{\frac{d_i}{d_{\min}^i}}\right) \cdot R,$$

$$\forall \|x' - x\| \leq R : \|f_i(x) - f_i(x')\| \leq (C \log d_{\max})^{i/2} \cdot \left(1 + \sqrt{\frac{d_i}{d_{\min}^i}}\right) \cdot R.$$

*Proof Sketch.* The proof of the first claim is nearly identical to that of Lemma 2.2.

For the second claim, we use a gridding argument with some subtleties. Concretely, we construct a new grid over the image of $\mathbb{B}(x, R)$ whenever the number of units in a hidden layer drops below all the previous layers in the network starting from the input layer. These layers are precisely defined by the indices $i_j$ in NN-DECOMP. We now establish the following claim inductively where we adopt the convention $i_0 = \ell + 1$.

**Claim 2.4.** Suppose for $j \geq 1$ and $\widetilde{R} \leq d_{\max} \cdot R$:

$$\forall y \in \mathbb{B}(x, R) : \|f_{i_j}(x) - f_{i_j}(y)\| \leq \widetilde{R}.$$

Then:

$$\forall i \in \{i_j + 1, \ldots, i_{j-1} - 1\}, y \in \mathbb{B}(x, R) : \|\widetilde{f}_i(x) - \widetilde{f}_i(y)\| \leq (C\ell \log d_{\max})^{(i - i_j)/2} \cdot \sqrt{\frac{d_i}{d_{i_j}}} \cdot \widetilde{R},$$

$$\forall y \in \mathbb{B}(x, R) : \|\widetilde{f}_{i_{j-1}}(x) - \widetilde{f}_{i_{j-1}}(y)\| \cdot \leq (C\ell \log d_{\max})^{(i_{j-1} - i_j)/2} \cdot \widetilde{R}$$

with probability at least $1 - \delta/8l$.

*Proof Sketch.* We start by constructing an $\varepsilon$-net [Ver18, Definition 4.2.1], $\mathcal{G}$, of $f_{i_j}(\mathbb{B}(x, R))$ with $\varepsilon = 1/(d_{\max}^\ell)^{32}$. Note that we may assume $|\mathcal{G}| \leq (10\widetilde{R}/\varepsilon)^{d_{i_j}}$. We will prove the statement on the grid and extend to the rest of the space. For layer $i + 1$, defining $\widetilde{x} = f_{i_j}(x)$, we have $\forall \widetilde{y} \in \mathcal{G}$:

$$\|\widetilde{f}_{i+1, i_j}(\widetilde{x}) - \widetilde{f}_{i+1, i_j}(\widetilde{y})\| = \|W_{i+1}(f_{i, i_j}(\widetilde{x}) - f_{i, i_j}(\widetilde{y}))\|$$

$$\leq \|f_{i, i_j}(\widetilde{x}) - f_{i, i_j}(\widetilde{y})\| \cdot \sqrt{\frac{d_{i+1}}{d_i}} \cdot \left(1 + \sqrt{\frac{\log 1/\delta'}{d_{i+1}}}\right)$$

with probability at least $1 - \delta'$ as before by [Theorem C.3](). By setting $\delta' = \delta/(16|\mathcal{G}|\ell^2)$ and noting that $d_i \geq d_{i_j}$, the conclusion holds for layer $i + 1 \leq i_j$ on $\mathcal{G}$ with probability at least $1 - \delta/(16\ell^2)$. By induction and the union bound, we get:

$$\forall i \in \{i_j + 1, \ldots, i_{j-1} - 1\}, \widetilde{y} \in \mathcal{G} : \|\widetilde{f}_{i, i_j}(\widetilde{y}) - \widetilde{f}_{i, i_j}(\widetilde{x})\| \leq (C\ell \log d_{\max})^{(i - i_j)/2} \cdot \sqrt{\frac{d_i}{d_{i_j}}} \cdot \widetilde{R}$$

$$\forall \widetilde{y} \in \mathcal{G} : \|\widetilde{f}_{i_{j-1}, i_j}(\widetilde{y}) - \widetilde{f}_{i_{j-1}, i_j}(\widetilde{x})\| \leq (C\ell \log d_{\max})^{(i_{j-1} - i_j)/2} \cdot \widetilde{R}$$

with probability at least $1 - \delta/(16\ell^2)$. To extend to all $y \in f_{i_j}(\mathbb{B}(x, R))$, we condition on the bound on $\|W_i\|$ given by [Lemma C.1]() for all $i \leq i_{j-1}$ and note that $\forall y \in f_{i_j}(\mathbb{B}(x, R))$, for $\widetilde{y} = \arg\min_{z \in \mathcal{G}} \|z - y\|$, and for $i_j + 1 \leq i < i_{j-1}$,

$$\|\widetilde{f}_{i, i_j}(\widetilde{x}) - \widetilde{f}_{i, i_j}(y)\| \leq \|\widetilde{f}_{i, i_j}(\widetilde{x}) - \widetilde{f}_{i, i_j}(\widetilde{y})\| + \|\widetilde{f}_{i, i_j}(y) - \widetilde{f}_{i, i_j}(\widetilde{y})\|$$

$$\leq \|\widetilde{f}_{i, i_j}(\widetilde{x}) - \widetilde{f}_{i, i_j}(\widetilde{y})\| + \|y - \widetilde{y}\| \prod_{k = i_j + 1}^{i} \|W_k\|$$

$$\leq \|\widetilde{f}_{i, i_j}(\widetilde{x}) - \widetilde{f}_{i, i_j}(\widetilde{y})\| + \varepsilon \prod_{k = i_j + 1}^{i} \left(C\sqrt{\frac{d_k}{d_{k-1}}}\right)$$

$$= \|\widetilde{f}_{i, i_j}(\widetilde{x}) - \widetilde{f}_{i, i_j}(\widetilde{y})\| + \varepsilon C^{i - i_j}\sqrt{\frac{d_i}{d_{i_j}}},$$

using that $d_i \geq d_{i_j}$. Similarly, for $i = i_{j-1}$, we have

$$\|\widetilde{f}_{i_{j-1}, i_j}(\widetilde{x}) - \widetilde{f}_{i_{j-1}, i_j}(y)\| \leq \|\widetilde{f}_{i_{j-1}, i_j}(\widetilde{x}) - \widetilde{f}_{i_{j-1}, i_j}(\widetilde{y})\| + \varepsilon C^{i_{j-1} - i_j}\sqrt{\frac{d_{i_{j-1}-1}}{d_{i_j}}}.$$

Our setting of $\varepsilon$ concludes the proof of the claim. □

An inductive application of [Claim 2.4](), a union bound and the observation that:

$$\|f_i(x) - f_i(y)\| \leq \|\widetilde{f}_i(x) - \widetilde{f}_i(y)\|$$

concludes the proof of the lemma. □

The following lemma shows that few neurons have inputs of small magnitude for network input $x$.

**Lemma 2.5.** *With probability at least $1 - \delta$, for all $i \in [\ell]$, we have:*

$$\#\left\{j : |\langle (W_{i+1})_j, f_i(x) \rangle| \geq \alpha_i \frac{\|f_i(x)\|}{\sqrt{d_i}}\right\} \geq \left(1 - 2\sqrt{\frac{2}{\pi}}\alpha_i\right) d_{i+1}.$$

*Proof.* Follows from the fact that $\Pr(|Z| \leq c) \leq c\sqrt{2/\pi}$ for $Z \sim N(0, 1)$, plus a simple application of Hoeffding's Inequality. □

## 2.4 Proving Local Gradient Smoothness

In this section, we show that the gradient is locally constant, and thus complete the proof of Theorem 1.1. In our proof, we will bound each of the terms in the expansion of the gradient differences (GD-DECOMP). First, we prove a structural lemma on the spectral norm of the matrices appearing in the right-hand-side of (GD-DECOMP), allowing us to ignore the portion of the network till the last-encountered bottleneck layer. Define, for all $i > j$, $M_{i,j}(y) = \prod_{k=i}^{j+1} D_k(\widetilde{f}_k(y))W_k$.

**Lemma 2.6.** *With probability at least $1 - \delta$ over the $\{W_k\}$, we have:*

$$\forall \|y - x\| \leq R, j \in [m-1] : \mathbb{P}_{\{D_1(\cdot),\ldots,D_\ell(\cdot)\}} \left\{ \left\|M_{i_j, i_{j+1}}(y)\right\| \leq (C \cdot \ell \cdot \log d_{\max})^{(i_j - i_{j+1})/2} \right\} = 1,$$

*where the probability is taken with respect to the random choices in the definition of the $D_k(\cdot)$.*

We provide the first part of the proof in full as the simplest application of ideas that find further application in the subsequent result establishing bounds on terms in (GD-DECOMP); see Appendix A.3.

*Proof Sketch.* To start, consider a fixed $j \in [m]$ and condition on the conclusion of Lemma 2.3 up to level $i_{j+1}$. Now, consider an $\varepsilon$-net of $f_{i_{j+1}}(\mathbb{B}(x, R))$, $\mathcal{G}$, with resolution $\varepsilon = 1/(d_{\max}^\ell)^{32}$. As before, $|\mathcal{G}| \leq (Cd_{\max})^{48\ell d_{i_{j+1}}}$ for some constant $C$. We additionally will consider subsets

$$\mathcal{S} = \left\{ (S_k)_{k=i_j-1}^{i_{j+1}+1} : S_k \subseteq [d_k], |S_k| \leq 4d_{i_{j+1}} \right\}.$$

Note that $|\mathcal{G}| \cdot |\mathcal{S}|^2 \leq (d_{\max})^{64\ell d_{i_{j+1}}}$. For $y \in \mathcal{G}, S^1, S^2 \in \mathcal{S}$, consider the following matrix:

$$M_{y,S^1,S^2}^{i_j,i_{j+1}} = \prod_{k=i_j}^{i_{j+1}+1} (D_k(\widetilde{f}_{k,i_{j+1}}(y))) + (D_{S_k^1} - D_{S_k^2}))W_k \text{ where } (D_S)_{i,j} = \begin{cases} 1, \text{if } i = j \text{ and } i \in S, \\ 0, \text{otherwise}. \end{cases}$$

We will bound the spectral norm of $M_{y,S^1,S^2}^{i_j,i_{j+1}}$. First, note that:

$$\left\|M_{y,S^1,S^2}^{i_j,i_{j+1}}\right\| \leq 2\left\|\widetilde{M}_{y,S^1,S^2}^{i_j,i_{j+1}}\right\| \text{ where } \widetilde{M}_{y,S^1,S^2}^{i_j,i_{j+1}} := W_{i_j} \prod_{k=i_j-1}^{i_{j+1}+1} (D_k(\widetilde{f}_{k,i_{j+1}}(y)) + (D_{S_k^1} - D_{S_k^2}))W_k$$

and observe that from Lemma 2.1:

$$\widetilde{M}_{y,S^1,S^2}^{i_j,i_{j+1}} \overset{d}{=} W_{i_j} \prod_{k=i_j-1}^{i_{j+1}+1} (D_k + (D_{S_k^1} - D_{S_k^2}))W_k) \text{ where } (D_k)_{i,j} = \begin{cases} 1 & \text{w.p } \frac{1}{2} \text{ if } i = j, \\ 0 & \text{otherwise}. \end{cases}$$

To bound the spectral norm, let $\mathcal{B}$ be a $1/3$-net of $\mathbb{S}^{d_{i_j}-1}$ and $v \in \mathcal{B}$. Applying Theorem C.3,

$$\left\|v^\top \widetilde{M}_{y,S^1,S^2}^{i_j,i_{j+1}}\right\| \leq \left\|v^\top W_{i_j} \prod_{k=i_j-1}^{i_{j+1}+2} (D_k + (D_{S_k^1} - D_{S_k^2}))W_k)\right\| \cdot \left(1 + \sqrt{\frac{\log 1/\delta'}{d_{i_{j+1}}}}\right)$$

$$\leq \prod_{k=i_j}^{i_{j+1}} \left(1 + \sqrt{\frac{\log 1/\delta'}{d_k}}\right)$$

with probability at least $\ell\delta'$. But setting $\delta' = \delta/(16\ell^4 \cdot |\mathcal{G}| \cdot |\mathcal{S}|^2)$ yields with probability at least $1 - \delta/16$:

$$\forall y \in \mathcal{G}, S_1, S_2 \in \mathcal{S} : \left\|\widetilde{M}_{y,S^1,S^2}^{i_j,i_{j+1}}\right\| \leq (C \cdot \ell \cdot \log d_{\max})^{(i_j - i_{j+1})/2}.$$

On the event in the conclusion of Lemma 2.3, we have that $f_i(y) \neq 0$ for all $i \in [\ell], y \in \mathcal{G}$ and therefore, we have by a union bound over the discrete set $\mathcal{G}$:

$$\forall y \in \mathcal{G}, k \in \{i_{j+1}, \ldots, i_j\}, m \in [d_k] : (\widetilde{f}_{k,i_j}(y))_m \neq 0$$

proving the lemma for $y \in \mathcal{G}$ as the activations are deterministic. For $y \notin \mathcal{G}$, the following claim concludes the proof of the lemma. The claim is essentially a generalization of [BCGdC21, Eq. (18)] and its proof is deferred to the appendix.

**Claim 2.7.** With probability at least $1 - \delta'/\ell^2$ over the $\{W_k\}$, we have for all $m \in \{i_{j+1} + 1, \ldots, i_j\}$ and $y \in f_{i_{j+1}}(\mathbb{B}(x, R))$:

$$\mathbb{P}_{\left\{D_k(\widetilde{f}_{k,i_{j+1}}(y)), D_k(\widetilde{f}_{k,i_{j+1}}(\widetilde{y}))\right\}} \left\{ \mathrm{Tr}|D_m(\widetilde{f}_{m,i_{j+1}}(y)) - D_m(\widetilde{f}_{m,i_{j+1}}(\widetilde{y}))| \le 4d_{i_{j+1}} \right\} = 1$$

where $\widetilde{y} = \arg\min_{z \in \mathcal{G}} \|z - y\|$, and the probability is taken with respect to the random choices in the definition of the $D_k(\cdot)$.

The previous claim along with our previously established bounds establish the lemma. $\qquad\square$

Our final technical result establishes the near-linearity of $f$ in a ball around $x$. The full proof is deferred to Appendix A.4 but in our proof sketch we identify sections which involve considerations unique to this lemma.

**Lemma 2.8.** *For some absolute constant $C$, with probability at least $1 - \delta$ over the $\{W_k\}$:*

$$\forall \|x - y\| \le R, j \in [\ell] : \mathbb{P}_{\left\{D_k(\widetilde{f}_k(x)), D_k(\widetilde{f}_k(y))\right\}} \left\{ \|\nabla f(x) - \nabla f(y)\| \le \left( \frac{C^\ell}{\log^\ell d_{\max}} \right) \right\} = 1,$$

*where the probability is taken with respect to the random choices in the definition of the $D_k(\cdot)$.*

*Proof Sketch.* Consider a fixed term from (GD-DECOMP); that is, consider:

$$\mathrm{Diff}_j(y) := W_{\ell+1} \left( \prod_{i=\ell}^{j+1} D_i(\widetilde{f}_i(x))W_i \right) \cdot (D_j(\widetilde{f}_j(y)) - D_j(\widetilde{f}_j(x)))W_j \cdot \left( \prod_{i=j-1}^{1} D_i(\widetilde{f}_i(y))W_i \right).$$

We will show with high probability that $\mathrm{Diff}_j(y)$ is small for all $y \in \mathbb{B}(x, R)$. This will then imply the lemma by a union bound and (GD-DECOMP). Let $k$ be such that $i_k = \arg\min_{m<j} d_m$. We will condition on the weights of the network up to layer $i_k$. Specifically, we will assume the conclusions of Lemmas 2.3 and 2.6 up to layer $i_k$. We may now focus our attention solely on the segment of the network beyond layer $i_k$ as a consequence of the following observation and Lemma 2.6:

$$\|\mathrm{Diff}_j(y)\| \le \|\mathrm{Diff}_{j,k}(x, y)\| \cdot \|M_{i_k,0}(y)\| \text{ where} \tag{3}$$

$$\mathrm{Diff}_{j,k}(x, y) := W_{\ell+1} \left( \prod_{i=\ell}^{j+1} D_i(\widetilde{f}_i(x))W_i \right) (D_j(\widetilde{f}_j(y)) - D_j(\widetilde{f}_j(x)))W_j \left( \prod_{i=j-1}^{i_k+1} D_i(\widetilde{f}_i(y))W_i \right)$$

We will show for all $y$ such that $\|y - x\| \le R$:

$$\mathbb{P}_{\left\{D_m(\widetilde{f}_m(x)), D_m(\widetilde{f}_m(y))\right\}} \left\{ \|\mathrm{Diff}_{j,k}(x, y)\| \ge \frac{C^\ell}{(\ell \log d_{\max})^{3\ell}} \right\} = 0$$

with probability at least $1 - \delta/(16\ell^2)$.

We have from Lemma 2.1 that the random vector $H$ defined below is spherically symmetric and satisfies $\|H\| \le 2^\ell$ with probability at least $1 - \delta/(16\ell^4)$:

$$W_{\ell+1} \prod_{i=\ell}^{j+1} D_i(\widetilde{f}_i(x))W_i \stackrel{d}{=} \widetilde{W}_{\ell+1} \prod_{i=\ell}^{j+1} D_i \widetilde{W}_i =: H.$$

As in the proof of Lemma 2.6, let $\mathcal{G}$ be an $\varepsilon$-net of $f_{i_k}(\mathbb{B}(x, R))$ with $\varepsilon$ as in the proof of Lemma 2.6. We now break into two cases depending on how $d_j$ compares to $d_{i_k}$ and handle them separately. At this point, we have effectively reduced the multi-layer proof to the problem of analyzing deviations of activations at a fixed layer. For the remaining proof, we generalize approaches in [DS20] for the small width case and [BCGdC21] for the large width case.

**Case 1:** $d_j \le d_{i_k}(\ell \log d_{\max})^{20\ell}$. In this case, the key observation already made in [DS20] is that under Lemma 2.3, the number of neurons that may actually differ at layer $j$ is at most $d_j/\mathrm{poly}(\ell \log d_{\max})^\ell$ between $y \in \mathbb{B}(x, R)$ and $x$. Since, $H$ is spherically distributed, $\|H(D_j(\widetilde{f}_j(x)) - D_j(\widetilde{f}_j(y)))\|$ is very small and consequently the whole term is small.

**Case 2:** $d_j \geq d_{i_k}(\ell \log d_{\max})^{20\ell}$. This case is analogous to [BCGdC21]. The proof is technically involved and requires careful analysis of the random vector $H(D_j(\widetilde{f}_j(x)) - D_j(\widetilde{f}_j(y)))W_j$, which is complicated in our setting due to the matrices preceeding it in (GD-DECOMP). It involves the distributional equivalence in Lemma 2.1.

A union bound and an application of the triangle inequality now imply the lemma. □

*Proof of Theorem 1.1.* On the intersection of the events in the conclusions of Lemmas 2.2, 2.3 and 2.8, we have that $\nabla f(x)$ is deterministic and furthermore, we have:

$$\|\nabla f(x)\| \geq \frac{1}{2^{\ell+1}}, \quad |f(x)| \leq c2^\ell \sqrt{\log 1/\delta}$$

$$\forall y \text{ s.t } \|y - x\| \leq R : \|\nabla f(x) - \nabla f(y)\| \leq \frac{1}{(\ell \log d_{\max})^\ell} = o(1).$$

Assume $f(x) > 0$ (the alternative is similar) and let $\eta = -\frac{2^\ell \log d \cdot \sqrt{\log 1/\delta}}{\|\nabla f(x)\|^2}$, we have for the point $x + \eta\nabla f(x)$, defining the function $g(t) = f(x + t\eta\nabla f(x))$:

$$f(x + \eta\nabla f(x)) = f(x) + \int_0^1 g'(t)dt = f(x) + \int_0^1 (\eta\nabla f(x))^\top \nabla f(x + t\eta\nabla f(x))dt$$

$$= f(x) - (1 - o(1))2^\ell \log d \sqrt{\log 1/\delta} \leq -f(x).$$

Our lower bounds on $\nabla f(x)$ ensure $\|\eta\nabla f(x)\| \leq R$, concluding the proof of the theorem. □

## 3 The Impact of Depth

Theorem 1.1 relies on the depth being constant. In this section, we show that some constraint on the depth is necessary in order to ensure the existence of adversarial examples. In particular, the following result gives an example of a sufficiently deep network for which, with high probability, the output of the network will have the same sign for all input patterns.

**Theorem 3.1.** *Fix a sufficiently large $d \in \mathbb{N}$, an $\ell \geq d^3$ and $(\ell d)^{20} \leq k \leq \exp(\sqrt{\ell})$, and consider the randomly initialized neural network (NN-DEF) with $d_1 = \cdots = d_\ell = k$. There is a universal constant $C$ such that with probability at least $0.9$,*

$$\forall x, y \in \mathbb{S}^{d-1} : \frac{|f(x) - f(y)|}{|f(x)|} \leq C\sqrt{\frac{\log d}{d}}.$$

The proof involves showing that the image of $\mathbb{S}^{d-1}$ is bounded away from $0$, and that the inner products between images of two input vectors throughout the network converge. The following lemma shows how the expected inner products evolve through the network. The lemma follows from computing a double integral; see [CS09, Eq. (6)]. The proof of Theorem 3.1 is in Appendix B.

**Lemma 3.2.** *Let $d \in \mathbb{N}$. Fix $x, y \in \mathbb{R}^d$ with $x, y \neq 0$. Then for $g \sim \mathcal{N}(0, I)$:*

$$\frac{\mathbb{E}\left[\max\{g^\top x, 0\} \max\{g^\top y, 0\}\right]}{\sqrt{\mathbb{E}(\max\{g^\top x, 0\})^2 \cdot \mathbb{E}(\max\{g^\top y, 0\})^2}} = \frac{\sin\theta}{\pi} + \left(1 - \frac{\theta}{\pi}\right)\cos\theta$$

*where $\theta = \arccos(x^\top y/(\|x\|\|y\|))$.*

## 4 Disclosure of Funding and Competing Interests

PB and YC gratefully acknowledge the support of the NSF through grants DMS-2023505 and DMS-2031883, the Simons Foundation through award #814639, and Microsoft through the BAIR Open Research Commons. We declare no competing interests.

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
