\left\{g^\top x, 0\right\} \max\left\{g^\top y, 0\right\}\right]}{\sqrt{\mathbb{E}(\max\left\{g^\top x, 0\right\})^2 \cdot \mathbb{E}(\max\left\{g^\top y, 0\right\})^2}} = \frac{\sin \theta}{\pi} + \left(1 - \frac{\theta}{\pi}\right)\cos \theta$$

*where $\theta = \arccos(x^\top y/(\|x\|\|y\|))$.*

## 4 Disclosure of Funding and Competing Interests

PB and YC gratefully acknowledge the support of the NSF through grants DMS-2023505 and DMS-2031883, the Simons Foundation through award #814639, and Microsoft through the BAIR Open Research Commons. We declare no competing interests.

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

## A  Deferred Proofs from Section 2

### A.1  Proof of Lemma 2.1

For the first claim, we introduce the following diagonal random signed matrices, $S_i$ for $i \in [m]$ with:

$$(S_i)_{j,j} = \pm 1 \text{ with equal probability}, \quad (S_i)_{j,k} = 0 \text{ if } j \neq k.$$

We prove the claim by induction on $m$. When $m = 1$, the claim is trivially true. For $m > 1$, we have:

$$W_m \prod_{j=m-1}^{1} (D_j(\widetilde{h}_j(x)) + B_j)W_j$$

$$\stackrel{d}{=} W_m \left( \prod_{j=m-1}^{2} (D_j(\widetilde{h}_j(x)) + B_j)W_j \right) (D_1(S_1\widetilde{h}_1(x)) + B_1)S_1W_1 \quad \left( S_1W_1 \stackrel{d}{=} W_1 \right)$$

$$\stackrel{d}{=} W_m \left( \prod_{j=m-1}^{2} (D_j + B_j)W_j \right) (D_1(S_1\widetilde{h}_1(x)) + B_1)S_1W_1 \qquad \text{(hypothesis for } m-1)$$

$$\stackrel{d}{=} W_m \left( \prod_{j=m-1}^{2} (D_j + B_j)W_j \right) S_1(D_1(S_1\widetilde{h}_1(x)) + B_1)S_1W_1 \qquad \left( W_2S_1 \stackrel{d}{=} W_2 \right)$$

$$= W_m \left( \prod_{j=m-1}^{2} (D_j + B_j)W_j \right) (D_1(S_1\widetilde{h}_1(x)) + B_1)W_1 \qquad (S_1B_1S_1 = B_1)$$

$$\stackrel{d}{=} W_m \left( \prod_{j=m-1}^{2} (D_j + B_j)W_j \right) (D_1 + B_1)W_1$$

$$\stackrel{d}{=} W_m \prod_{j=m-1}^{1} (D_j + B_j)W_j.$$

Similarly, for the second claim, we have:

$$\left\| \prod_{j=m}^{1} (D_j(\widetilde{h}_j(x)) + B_j)W_j \right\|$$

$$\stackrel{d}{=} \left\| (D_m(S_m\widetilde{h}_m(x)) + B_m)S_mW_m \prod_{j=m-1}^{1} (D_j(\widetilde{h}_j(x)) + B_j)W_j \right\|$$

$$\stackrel{d}{=} \left\| S_m(D_m(S_m\widetilde{h}_m(x)) + B_m)S_mW_m \prod_{j=m-1}^{1} (D_j(\widetilde{h}_j(x)) + B_j)W_j \right\|$$

$$\stackrel{d}{=} \left\| (D_m(S_m\widetilde{h}_m(x)) + B_m)W_m \prod_{j=m-1}^{1} (D_j(\widetilde{h}_j(x)) + B_j)W_j \right\|$$

$$\stackrel{d}{=} \left\| (D_m + B_m)W_m \prod_{j=m-1}^{1} (D_j(\widetilde{h}_j(x)) + B_j)W_j \right\|$$

$$\stackrel{d}{=} \left\| \prod_{j=m}^{1} (D_j + B_j)W_j \right\|$$

$\square$

## A.2    Proof of Lemma 2.3

We start with the first claim of the lemma. As in Lemma 2.2, we note from Lemma 2.1:

$$f_i(x) = \left( \prod_{j=i}^{1} D_j(\widetilde{f}_j(x)) W_j \right) x$$

$$\left\| \prod_{j=i}^{1} D_j(\widetilde{f}_j(x)) W_j \right\| \overset{d}{=} \left\| \prod_{j=i}^{1} D_j \widetilde{W}_j \right\| \text{ where}$$

$$(D_i)_{j,k} = \begin{cases} 1 & \text{w.p } 1/2 \text{ if } j = k \\ 0 & \text{otherwise} \end{cases} \text{ and } \{W_i\}_{i=1}^{\ell+1} \overset{d}{=} \{\widetilde{W}_i\}_{i=1}^{\ell+1}.$$

Therefore, it suffices to analyze the distribution of $\prod_{j=i}^{1} D_j \widetilde{W}_j x$. Again, we condition on the following event:

$$\forall i \in [\ell] : \operatorname{Tr} D_i \geq \frac{d_i}{3}$$

which occurs with probability at least $1 - \delta/8$. As in the proof of Lemma 2.2, we observe the following distributional equivalence, when we condition on the $D_j$:

$$\forall i \in [\ell] : \left\| \prod_{j=i}^{1} D_j \widetilde{W}_j x \right\| \overset{d}{=} \left\| \prod_{j=i}^{1} W_j^\dagger x \right\| \text{ where}$$

$$\forall j \in \{2, \dots, i\}, W_j^\dagger \in \mathbb{R}^{\operatorname{Tr} D_j \times \operatorname{Tr} D_{j-1}}, W_1^\dagger \in \mathbb{R}^{\operatorname{Tr} D_1 \times d}$$

$$\forall j \in \{2, \dots, i\}, (W_j^\dagger)_{k,:} \sim \mathcal{N}(0, I/d_{j-1}), (W_1^\dagger)_{k,:} \sim \mathcal{N}(0, I/d).$$

A recursive application of Theorem C.3 as in the proof of Lemma 2.2 yields:

$$\left\| \prod_{j=i}^{1} D_j \widetilde{W}_j x \right\| \geq \|x\| \prod_{j=i}^{1} \frac{1}{2} \sqrt{\frac{d_j}{d_{j-1}}}$$

with probability at least $1 - \delta/16\ell^2$. A union bound over the $j$ layers concludes the proof of the first claim of the lemma.

We will establish the second claim through a gridding based argument with some subtleties. Namely,

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

$$\frac{\log 1/\delta'}{d_k} \le \frac{C\ell d_{i_{j+1}}\log d_{\max}}{d_k} + \frac{\log 1/\delta}{d_k}$$

$$\le C'\ell \log d_{\max}$$

because $d_k \ge d_{i_{j+1}}$ and by the assumption on $\delta$. Hence, with probability at least $1 - \delta/16$:

$$\forall y \in \mathcal{G}, S_1, S_2 \in \mathcal{S} : \left\|\widetilde{M}_{y,S^1,S^2}^{i_j,i_{j+1}}\right\| \le (C \cdot \ell \cdot \log d_{\max})^{(i_j-i_{j+1})/2}.$$

On the event in the conclusion of Lemma 2.3, we have that for all $y \in \mathbb{B}(x, R)$ and $i \in [\ell]$, $f_i(y) \ne 0$, and therefore we have by a union bound over the discrete set $\mathcal{G}$ in an event of probability 1,

$$\forall y \in \mathcal{G}, k \in \{i_{j+1}, \ldots, i_j\}, m \in [d_k] : (\widetilde{f}_{k,i_j}(y))_m \ne 0.$$

We now show a basic structural claim of how activation patterns differ across the various layers between a point and its closest approximation in $\mathcal{G}$:

**Claim A.2.** With probability at least $1 - \delta'/\ell^2$ over the $W_k$, we have for all $m \in \{i_{j+1} + 1, \ldots, i_j\}$ and $y \in f_{i_{j+1}}(\mathbb{B}(x, R))$:

$$\mathbb{P}_{\left\{D_n(\widetilde{f}_{n,i_{j+1}}(y)), D_n(\widetilde{f}_{n,i_{j+1}}(\widetilde{y}))\right\}}\left\{\text{Tr}|D_m(\widetilde{f}_{m,i_{j+1}}(y)) - D_m(\widetilde{f}_{m,i_{j+1}}(\widetilde{y}))| \le 4d_{i_{j+1}}\right\} = 1,$$

where $\widetilde{y} = \arg\min_{z \in \mathcal{G}}\|z - y\|$.

*Proof.* We start by conditioning on the conclusion of Lemma C.1 (with $A = \sqrt{d_{k-1}}W_k$) up to layer $i_{j+1}$. Then:

$$\forall m \in \{i_{j+1}+1, \cdots, i_j\}, y \in f_{i_{j+1}}(\mathbb{B}(x, R)) : \|\widetilde{f}_{m,i_{j+1}}(y) - \widetilde{f}_{m,i_{j+1}}(\widetilde{y})\| \le \varepsilon\cdot(Cd_{\max})^{(m-i_{j+1})/2}.$$

Now, let $m \in \{i_{j+1} + 1, \cdots, i_j\}$ and $y \in \mathcal{G}$. We now operate on the conclusion of Lemma 2.3 up to layer $m - 1$ and hence, $(\widetilde{f}_{m,i_{j+1}}(y))_n \ne 0$ for all $n$. Now, we have:

$$\max_{z \text{ s.t } \|z-y\|\le\varepsilon} \#\{i : \text{sign}(\langle(W_m)_{i,:}, f_{m-1,i_{j+1}}(y)\rangle) \ne \text{sign}(\langle(W_m)_{i,:}, f_{m-1,i_{j+1}}(z)\rangle)\}$$

$$\le \#\{i : \exists z \text{ s.t } \|z-y\| \le \varepsilon \text{ and } \text{sign}(\langle(W_m)_{i,:}, f_{m-1,i_{j+1}}(y)\rangle) \ne \text{sign}(\langle(W_m)_{i,:}, f_{m-1,i_{j+1}}(z)\rangle)\}.$$

Defining $w_i := (W_m)_{i,:}$ and

$$W_i(y) := \mathbf{1}\{\exists z \text{ s.t } \|z-y\| \le \varepsilon \text{ and } \operatorname{sign}(\langle (W_m)_{i,:}, f_{m-1,i_{j+1}}(y)\rangle) \neq \operatorname{sign}(\langle (W_m)_{i,:}, f_{m-1,i_{j+1}}(z)\rangle)\},$$

we get:

$$\mathbb{P}\{W_i(y) = 1\} \le \mathbb{P}\left\{(Cd_{\max})^{(m-1-i_{j+1})/2} \cdot \varepsilon \cdot \|w_i\| \ge |\langle w_i, y\rangle|\right\}$$

$$\le \mathbb{P}\{\|w_i\| \ge 2\} + \mathbb{P}\left\{2\varepsilon(Cd_{\max})^{(m-1-i_{j+1})/2} \ge |\langle w_i, y\rangle|\right\} \le \varepsilon \cdot (Cd_{\max})^{(m-i_{j+1})/2}.$$

Therefore, we have:

$$\mathbb{P}\left\{\sum_{i=1}^{d_m} W_i(y) \ge 4d_{i_{j+1}}\right\} \le d_m^{4d_{i_{j+1}}} \cdot \left(\varepsilon(Cd_{\max})^{(m-i_{j+1})/2}\right)^{4d_{i_{j+1}}}$$

and hence by the union bound and our setting of $\varepsilon$ and bound on $\mathcal{G}$:

$$\mathbb{P}\left\{\exists y \in \mathcal{G} : \sum_{i=1}^{d_m} W_i(y) \ge 4d_{i_{j+1}}\right\} \le (Cd_{\max})^{60\ell d_{i_{j+1}}} \cdot \varepsilon^{4d_{i_{j+1}}} \le \frac{\delta'}{16\ell^4}.$$

$\square$

The claim concludes the proof of the lemma. $\square$

## A.4  Proof of Lemma 2.8

Consider a fixed term in the decomposition of the gradient difference from GD-DECOMP; that is, consider the random vector-valued function:

$$\operatorname{Diff}_j(y) := W_{\ell+1}\left(\prod_{i=\ell}^{j+1} D_i(\widetilde{f}_i(x))W_i\right) \cdot (D_j(\widetilde{f}_j(y)) - D_j(\widetilde{f}_j(x)))W_j \cdot \left(\prod_{i=j-1}^{1} D_i(\widetilde{f}_i(y))W_i\right).$$

We will show with high probability that $\operatorname{Diff}_j(y) = o(1)$ with high probability for all $\|x - y\| \le R$. This will then imply the lemma by a union bound and GD-DECOMP. Let $k$ be such that $i_k = \arg\min_{m<j} d_m$. We will condition on the weights of the network up to layer $i_k$. Specifically, we will assume the conclusions of Lemmas 2.3 and 2.6 up to layer $i_k$. We may now focus our attention solely on the segment of the network beyond layer $i_k$ as a consequence of the following observations:

$$\|\operatorname{Diff}_j(y)\| \le \|\operatorname{Diff}_{j,k}(x,y)\| \cdot \|M_{i_k,0}(y)\| \text{ where} \tag{4}$$

$$\operatorname{Diff}_{j,k}(x,y) := W_{\ell+1}\left(\prod_{i=\ell}^{j+1} D_i(\widetilde{f}_i(x))W_i\right)(D_j(\widetilde{f}_j(y)) - D_j(\widetilde{f}_j(x)))W_j\left(\prod_{i=j-1}^{i_k+1} D_i(\widetilde{f}_i(y))W_i\right)$$

We will show for all $y$ such that $\|y - x\| \le R$:

$$\mathbb{P}_{\left\{D_m(\widetilde{f}_m(x)), D_m(\widetilde{f}_m(y))\right\}}\left\{\|\operatorname{Diff}_{j,k}(x,y)\| \ge \frac{C^\ell}{(\ell \log d_{\max})^{3\ell}}\right\} = 0 \tag{5}$$

with probability at least $1 - \delta/(16\ell^2)$.

Observe now that from Lemma 2.1:

$$W_{\ell+1}\prod_{i=\ell}^{j+1} D_i(\widetilde{f}_i(x))W_i \stackrel{d}{=} \widetilde{W}_{\ell+1}\prod_{i=\ell}^{j+1} D_i\widetilde{W}_i =: H \text{ where}$$

$$(D_i)_{m,n} = \begin{cases} 1, & \text{with probability } 1/2 \text{ if } m = n \\ 0, & \text{otherwise} \end{cases}, \{W_i\}_{i=j+1}^{\ell+1} \stackrel{d}{=} \{\widetilde{W}_i\}_{i=j+1}^{\ell+1}$$

Therefore, $H$ is a spherically-symmetric random vector and we condition on the following high probability bound on its length:

$$\|H\| \leq \left\| \widetilde{W}_{\ell+1} \prod_{i=\ell}^{j+2} D_i \widetilde{W}_i \right\| \cdot \left( 1 + 2\sqrt{\frac{\log \ell/\delta'}{d_j}} \right) \leq (2)^{\ell+1-j} \tag{6}$$

with probability at least $1 - \delta'$ and setting $\delta' = \delta/(64\ell^2)$. Observe that the distribution of $H$ remains spherically symmetric even after conditioning on the above event. As in the proof of Lemma 2.6, let $\mathcal{G}$ be an $\varepsilon$-net of $f_{i_k}(\mathbb{B}(x, R))$ with $\varepsilon$ as in the proof of Lemma 2.6.

**Claim A.3.** We have for all $m \in \{i_k + 1, \ldots, i_{k-1}\}, y \in f_{i_k}(\mathbb{B}(x, R))$:

$$\mathbb{P}_{\{D_m(\widetilde{f}_{m,i_k}(y)), D_m(\widetilde{f}_{m,i_k}(\widetilde{y}))\}} \left\{ \mathrm{Tr}|D_m(\widetilde{f}_{m,i_k}(y)) - D_m(\widetilde{f}_{m,i_k}(\widetilde{y}))| \leq 4d_{i_k} \right\} = 1$$

where $\widetilde{y} = \arg\min_{z \in \mathcal{G}} \|z - y\|$ with probability at least $1 - \delta'/\ell^2$.

*Proof.* The proof is identical to the proof of Claim A.2. $\qquad \square$

We now break into two cases depending on how $d_j$ compares to $d_{i_k}$ and handle them separately.

**Case 1:** $d_j \leq d_{i_k}(\ell \log d_{\max})^{20\ell}$. In this case, define the sets $\mathcal{S}, \mathcal{Q}$ as follows:

$$\mathcal{S} := \left\{ (S_m)_{m=i_k+1}^{j-1} : S_m \subseteq [d_m], \ |S_m| \leq 4d_{i_k} \right\},$$

$$\mathcal{Q} := \left\{ Q : Q \subset [d_j], \ |Q| \leq \frac{d_j}{(\ell \log d_{\max})^{60\ell}} \right\}.$$

Now, for $Q^1, Q^2 \in \mathcal{Q}, y \in \mathcal{G}, S^1, S^2 \in \mathcal{S}$, define the random vector:

$$V_{y,Q^1,Q^2,S^1,S^2}^{j,k} = H(D_{Q^1} - D_{Q^2})W_j \prod_{m=j-1}^{i_k+1} (D_m(\widetilde{f}_{m,i_k}(y)) + (D_{S_m^1} - D_{S_m^2}))W_m.$$

Note that we have from Lemma 2.1:

$$V_{y,Q^1,Q^2,S^1,S^2}^{j,k} \overset{d}{=} H(D_{Q^1} - D_{Q^2})W_j \prod_{m=j-1}^{i_k+1} (D_m + (D_{S_m^1} - D_{S_m^2}))W_m \text{ where}$$

$$(D_m)_{i,j} = \begin{cases} 1, & \text{w.p } 1/2 \text{ if } i = j \\ 0, & \text{otherwise} \end{cases}.$$

Note that $|\mathcal{Q}|^2 \cdot |\mathcal{G}| \cdot |S|^2 \leq (d_{\max})^{64\ell d_{i_k}}$, because of the condition on $d_j$ in this case. We now get:

$$\left\| V_{y,Q^1,Q^2,S_1,S_2}^{j,k} \right\|$$

$$\leq \left\| H(D_{Q^1} - D_{Q^2})W_j \prod_{m=j-1}^{i_k+2} (D_m + (D_{S_m^1} - D_{S_m^2}))W_m \right\| \cdot \left( 1 + \sqrt{\frac{\log 1/\delta^\dagger}{d_{i_k}}} \right)$$

$$\leq \|H(D_{Q^1} - D_{Q^2})\| \cdot \prod_{m=i_k+1}^{j} \left( 1 + \sqrt{\frac{\log 1/\delta^\dagger}{d_{m-1}}} \right)$$

$$\leq \|H(D_{Q^1} - D_{Q^2})\| \cdot (C\ell \log d_{\max})^{(j-i_k)/2}$$

where the final inequality follows from the fact that $d_m \geq d_{i_k}$ and by setting $\delta^\dagger = \delta/(32 \cdot \ell^4 \cdot |\mathcal{Q}|^2 \cdot |\mathcal{G}| \cdot |\mathcal{S}|^2)$. A union bound now implies that the previous conclusion holds for all $Q^1, Q^2 \in \mathcal{Q}, y \in \mathcal{G}, S^1, S^2 \in \mathcal{S}$. For the first term, we use the trivial bound:

$$\|H(D_{Q^1} - D_{Q^2})\| \leq \sqrt{|Q^1| + |Q^2|} \cdot \|H\|_\infty$$

$$\leq 10 \cdot \sqrt{\frac{d_j}{(\ell \log d_{\max})^{60\ell}}} \cdot \sqrt{\frac{\|H\|^2}{d_j} \cdot (\log d_{\max} + \log \ell/\delta)}.$$

with probability at least $1 - \delta/(32 \cdot \ell^4)$ from Lemma C.6 with $M$ set to the standard basis vectors.

To conclude the proof, we get from Lemmas 2.3 and 2.5, for all $y$ s.t $\|y - x\| \leq R$:

$$\mathrm{Tr}|D_j(\widetilde{f}_j(x)) - D_j(\widetilde{f}_j(y))|$$

$$\leq \# \left\{ i : |\langle (W_j)_i, f_{j-1}(x) \rangle| \leq \frac{\|f_{j-1}(x)\|}{4\sqrt{d_{j-1}} \cdot (\ell \log d_{\max})^{75\ell}} \right\}$$

$$+ \frac{\|\widetilde{f}_j(x) - \widetilde{f}_j(y)\|^2}{\|f_{j-1}(x)\|^2} \cdot 16 \cdot d_{j-1} \cdot (\ell \log d_{\max})^{150\ell}$$

$$\leq \frac{d_j}{2(\ell \log d_{\max})^{75\ell}} + d_j \left( \frac{R^2}{d_{\min}} \right) \frac{(C\ell \log d_{\max})^{\ell}}{d_{j-1}} \cdot 16 d_{j-1} (\ell \log d_{\max})^{150\ell} \leq \frac{d_j}{(\ell \log d_{\max})^{60\ell}}.$$

where the first inequality follows from the fact that for all $t > 0$:

$$|T| \leq \frac{\|\widetilde{f}_j(x) - \widetilde{f}_j(y)\|^2}{t^2} \text{ where}$$

$$T := \{i : |\langle (W_j)_i, f_{j-1}(x) \rangle| \geq t \text{ and } \mathrm{sign}(\langle (W_j)_i, f_{j-1}(x) \rangle) \neq \mathrm{sign}(\langle (W_j)_i, f_{j-1}(y) \rangle)\}.$$

Hence, we get with probability at least $1 - \delta/(16\ell^4)$:

$$\forall \|y - x\| \leq R : \mathrm{Tr}|D_j(\widetilde{f}_j(y)) - D_j(\widetilde{f}_j(x))| \leq \frac{d_j}{(\ell \log d_{\max})^{60\ell}},$$

$$\forall \|y - x\| \leq R, \forall m \in \{i_k + 1, \dots, j - 1\} : \mathrm{Tr}|D_m(\widetilde{f}_m(y)) - D_m(\widetilde{f}_{m,i_k}(\widetilde{y}))| \leq 4 d_{i_k},$$

where $\widetilde{y} = \arg\min_{z \in \mathcal{G}} \|z - f_{i_k}(y)\|$.

To conclude (5), note that for all $y \in \mathcal{G}$ we have that $(\widetilde{f}_{m,i_k}(y))_n \neq 0$ almost surely on Lemma 2.3. Hence, all the $D_m(\widetilde{f}_{m,i_k}(y))$ are deterministic on Lemma 2.3 and similarly for $x$. This immediately yields (5) for $y \in \mathcal{G}$. For $y \notin \mathcal{G}$, the conclusion follows from the previous discussion and Claim A.3.

**Case 2:** $d_j \geq d_{i_k}(\ell \log d_{\max})^{20\ell}$. As in the previous case, we start by defining the sets $\mathcal{S}$:

$$\mathcal{S} = \left\{ (S_m)_{m=i_k+1}^{j-1} : S_m \subseteq [d_m], |S_m| \leq 4 \cdot d_{i_k} \right\}.$$

Now, for $y \in \mathcal{G}, S^1, S^2 \in \mathcal{S}$, consider the random matrix $M_{y,S^1,S^2}^{j,k}$ and an application of Lemma 2.1:

$$M_{y,S^1,S^2}^{j,k} := W_{j-1} \prod_{m=j-2}^{i_k+1} (D_m(\widetilde{f}_m(y)) + (D_{S_m^1} - D_{S_m^2})) W_m$$

$$M_{y,S^1,S^2}^{j,k} \overset{d}{=} \widetilde{W}_{j-1} \prod_{m=j-2}^{i_k+1} (D_m + (D_{S_m^1} - D_{S_m^2})) \widetilde{W}_m.$$

We will bound the spectral norm of $M_{y,S^1,S^2}^{j,k}$ for all $y, S^1, S^2$ with high probability as follows. Let $\mathcal{V}$ be a $1/9$-grid of $\mathbb{S}^{d_{i_k}}$ and $v \in \mathcal{V}$. We have:

$$\left\| M_{y,S^1,S^2}^{j,k} v \right\| \leq \sqrt{\frac{d_{j-1}}{d_{j-2}}} \cdot \left( 1 + \sqrt{\frac{\log 1/\delta^\dagger}{d_{j-1}}} \right) \cdot \left\| \prod_{m=j-2}^{i_k+1} (D_m + (D_{S_m^1} - D_{S_m^2})) \widetilde{W}_m v \right\|$$

$$\leq 2^{(j-i_k+1)} \sqrt{\frac{d_{j-1}}{d_{i_k}}} \cdot \prod_{m=i_k+1}^{j-1} \left( 1 + \sqrt{\frac{\log 1/\delta^\dagger}{d_m}} \right)$$

$$\leq \sqrt{\frac{d_{j-1}}{d_{i_k}}} \cdot (C\ell \log d_{\max})^{(j-i_k+1)/2} \tag{7}$$

where the final inequality follows by the fact that $d_m \geq d_{i_k}$ and by setting $\delta^\dagger = \delta/(32 \cdot \ell \cdot |\mathcal{G}| \cdot |\mathcal{S}|^2 \cdot \mathcal{V})$. By a union bound, the bound on the spectral norms of $M_{y,S^1,S^2}^{j,k}$ follows. We now condition on this event and the conclusion of Lemma 2.3 up to layer $j-1$ for the rest of the proof, which as before implies that $M_{y,S^1,S^2}^{j,k}$ and $D_{j-1}(\widetilde{f}_{j-1,i_k}(y))$ are no longer random.

Let $\widetilde{M}_{y,S^1,S^2}^{j,k} = D_{j-1}(\widetilde{f}_{j-1,i_k}(y)) \cdot M_{y,S^1,S^2}^{j,k}$ and for $y \in \mathcal{G}$, let $\widetilde{y} = f_{j-1,i_k}(y)$ and $\widetilde{x} = f_{j-1}(x)$. We have by an application of Lemma A.4 and a union bound over all $y \in \mathcal{G}, S^1, S^2 \in \mathcal{S}$:

$$\forall y \in \mathcal{G}, S^1, S^2 \in \mathcal{S} : \left\| H^\top \left( D_j(\widetilde{f}_{j,i_k}(y)) - D_j(\widetilde{f}_j(x)) \right) W_j \widetilde{M}_{y,S^1,S^2}^{j,k} \right\| \leq \left( \frac{C}{\ell \log d_{\max}} \right)^{3\ell}$$

with probability at least $1 - \delta'/\ell^2$ by recalling that $d_j \geq d_{i_k}(\ell \log d_{\max})^{20\ell}$ in this case.

We now additionally condition on $\|H\|_\infty$. We have as a consequence of Lemma C.6 that:

$$\|H\|_\infty \leq 4 \cdot 2^{\ell+1-j} \cdot \sqrt{\frac{\log d_j + \log 1/\delta^\dagger}{d_j}}$$

with probability at least $1 - \delta^\dagger$. We condition on this event and proceed as follows. Let $T \subset [d_j]$ such that $|T| \leq 4 \cdot d_{i_k}$ and $y \in \mathcal{G}, S^1, S^2, \in \mathcal{S}$ and we observe:

$$\left\| H^\top D_T \widetilde{W}_j \widetilde{M}_{y,S^1,S^2}^{j,k} \right\| \leq \left\| H^\top D_T \widetilde{W}_j U_{y,S^1,S^2}^{j,k} \right\| \cdot \left\| \widetilde{M}_{y,S^1,S^2}^{j,k} \right\|$$

where $U_{y,S^1,S^2}^{j,k}$ are the left singular vectors of $\widetilde{M}_{y,S^1,S^2}^{j,k}$ and observe:

$$\mathbb{P}\left\{ \left\| H^\top D_T \widetilde{W}_j U_{y,S^1,S^2}^{j,k} \right\| \geq t \right\} = \mathbb{P}_{\widetilde{Z} \sim \mathcal{N}\left(0, \|HD_T\|^2 \cdot \frac{I}{d_j}\right)} \left\{ \|\widetilde{Z}\| \geq t \right\}$$

$$\leq \mathbb{P}_{Z \sim \mathcal{N}\left(0, \|H\|_\infty^2 \cdot |T| \cdot \frac{I}{d_j}\right)} \left\{ \|Z\| \geq t \right\}$$

to get that (conditioned on $H$ and noting that $\mathrm{rank}(U_{y,S^1,S^2}^{j,k}) \leq d_{i_k}$), with probability $1 - \delta^\ddagger$:

$$\left\| H^\top D_T U_{y,S^1,S^2}^{j,k} \right\| \leq \|H\|_\infty \cdot \sqrt{|T|} \cdot \sqrt{\frac{d_{i_k}}{d_j}} \cdot \left( 1 + \sqrt{\frac{\log 1/\delta^\ddagger}{d_{i_k}}} \right)$$

By setting $\delta^\ddagger = \delta/(128\ell^2 d_j^{4d_{i_k}} |\mathcal{S}|^2 |\mathcal{G}|)$ and our bounds on $d_j, \|H\|_\infty$ yield:

$$\forall T \subset [d_j] \text{ s.t } |T| \leq 4d_{i_k}, y \in \mathcal{G}, S^1, S^2 \in \mathcal{S} : \left\| H^\top D_T U_{y,S^1,S^2}^{j,k} \right\| \leq \frac{1}{(\ell \log d_{\max})^{4\ell}}$$

with probability at least $1 - \delta/(64\ell^2)$ again by recalling $d_j \geq d_{i_k}(\ell \log d_{\max})^{20\ell}$.

To conclude our proof of (5), we proceed similarly to the previous case. As before, for all $y \in \mathcal{G}$ we have that $(\widetilde{f}_{m,i_k}(y))_n \neq 0$ almost surely on Lemma 2.3 and similarly for $x$. Hence, all the $D_m(\widetilde{f}_{m,i_k}(y))$ are deterministic on Lemma 2.3. This immediately yields (5) for $y \in \mathcal{G}$. For $y \notin \mathcal{G}$, the conclusion follows from the previous discussion and Claim A.3.

A union bound over all $j \in [\ell]$, an application of the triangle inequality with GD-DECOMP and (4) with Lemma 2.6 conclude the proof of the lemma. $\square$

**Lemma A.4.** *Let $L, R, r > 0, k, m, n \in \mathbb{N}, M \in \mathbb{R}^{m \times n}$ such that $m \geq n$ and $R > r$. Furthermore, suppose $W \in \mathbb{R}^{k \times m}$ and $H$ are distributed as follows:*

$$W = \begin{bmatrix} w_1^\top \\ w_2^\top \\ \vdots \\ w_k^\top \end{bmatrix} \text{ with } w_i \overset{i.i.d}{\sim} \mathcal{N}(0, I/m), \text{ and } H \sim \mathrm{Unif}(L \cdot \mathbb{S}^{k-1}).$$

*Furthermore, suppose $x \in \mathbb{R}^m$ satisfies $\|x\| \geq R$ and $y$ be such that $\|y - x\| \leq r$. Then:*

$$\mathbb{P}\left\{ \|H^\top DWM\| \geq 1024L\|M\| \left( \sqrt{\frac{3r}{R} \log R/r} \left( \frac{n + \log 1/\delta^\dagger}{m} \right) + \left( \frac{n + \log 1/\delta^\dagger}{\sqrt{km}} \right) \right) \right\} \leq \delta^\dagger$$

$$\text{where } D = D_x - D_y \text{ with } (D_x)_{i,j} = \begin{cases} 1, & \textit{if } i = j \textit{ and } w_i^\top x > 0 \\ 1, & \textit{w.p } 1/2 \textit{ if } i = j \textit{ and } w_i^\top x = 0 \\ 0, & \textit{otherwise} \end{cases}.$$

*Proof.* We may discard the cases where $w_i^\top x, w_i^\top y = 0$ as these form a measure 0 set. Now, it suffices to analyze the random variable:

$$\widetilde{H}^\top DWM \text{ for } \widetilde{H} \sim \mathcal{N}(0, 2L^2 \cdot I/k).$$

Let $V$ denote the two-dimensional subspace of $\mathbb{R}^m$ containing $x, y$. We now decompose the norm as follows:

$$\|\widetilde{H}^\top DWM\| \le \|\widetilde{H}^\top DW\mathcal{P}_V M\| + \|\widetilde{H}^\top DW\mathcal{P}_V^\perp M\| \le \|\widetilde{H}^\top DW\mathcal{P}_V\| \cdot \|M\| + \|\widetilde{H}^\top DW\mathcal{P}_V^\perp M\|. \tag{8}$$

To apply Bernstein's inequality, we first expand on the first term:

$$\left(\widetilde{H}^\top DW\mathcal{P}_V\right)^\top = \sum_{i=1}^k \widetilde{H}_i D_{i,i} \mathcal{P}_V w_i.$$

Letting $Z_i = \widetilde{H}_i D_{i,i} \mathcal{P}_V w_i$, we note that $\mathbb{E}[Z_i] = 0$ and bound its even moments as follows, in an orthonormal basis $\{v_1, v_2\}$ for $V$:

$$\mathbb{E}\left[\langle v_j, Z_i\rangle^2\right] = \mathbb{E}\left[\widetilde{H}_i^2 \cdot D_{i,i}^2 \cdot \langle v_j, w_i\rangle^2\right]$$

$$\le \frac{2L^2}{k} \cdot \mathbb{E}\left[\|\mathcal{P}_V w_i\|^2\right] \cdot \mathbb{P}\{D_{i,i} \ne 0\} = \frac{2L^2}{k} \cdot \frac{2}{m} \cdot \mathbb{P}\{D_{i,i} \ne 0\}$$

$$\mathbb{E}\left[\langle v_j, Z_i\rangle^\ell\right] = \mathbb{E}\left[\widetilde{H}_i^\ell \cdot D_{i,i}^\ell \cdot \langle v_j, w_i\rangle^\ell\right] \le (\ell-1)!! \left(\frac{2L^2}{k}\right)^{\ell/2} \cdot \mathbb{E}\left[\|\mathcal{P}_V w_i\|^\ell\right] \cdot \mathbb{P}\{D_{i,i} \ne 0\}$$

$$\le (\ell-1)!! \left(\frac{2L^2}{k}\right)^{\ell/2} \cdot \left(2^{\ell/2} \cdot \mathbb{E}\left[\langle w_i, v_1\rangle^\ell + \langle w_i, v_2\rangle^\ell\right]\right) \cdot \mathbb{P}\{D_{i,i} \ne 0\}$$

$$= 2 \cdot (\ell-1)!! \left(\frac{4L^2}{k}\right)^{\ell/2} \cdot \mathbb{E}\left[\langle w_i, v_1\rangle^\ell\right] \cdot \mathbb{P}\{D_{i,i} \ne 0\}$$

$$= 2 \cdot ((\ell-1)!!)^2 \left(\frac{4L^2}{km}\right)^{\ell/2} \cdot \mathbb{P}\{D_{i,i} \ne 0\} \le 2 \cdot \ell! \cdot \left(\frac{8L^2}{km}\right)^{\ell/2} \cdot \mathbb{P}\{D_{i,i} \ne 0\}.$$

For odd $\ell \ge 3$ we have by similar manipulations:

$$\mathbb{E}\left[|\langle v_j, Z_i\rangle|^\ell\right] = \mathbb{E}\left[\left|\widetilde{H}_i^\ell \cdot D_{i,i}^\ell \cdot \langle v_j, w_i\rangle^\ell\right|\right] \le \mathbb{P}\{D_{i,i} \ne 0\} \cdot \mathbb{E}\left[|\widetilde{H}_i^\ell \cdot \|\mathcal{P}_V w_i\|^\ell|\right]$$

$$\le \mathbb{P}\{D_{i,i} \ne 0\} \cdot \sqrt{\mathbb{E}\left[\left(\widetilde{H}_i \cdot \|\mathcal{P}_V w_i\|\right)^{2\ell}\right]} \le \mathbb{P}\{D_{i,i} \ne 0\} \cdot \sqrt{2 \cdot (2\ell)! \cdot \left(\frac{8L^2}{km}\right)^\ell}$$

$$\le \mathbb{P}\{D_{i,i} \ne 0\} \cdot \sqrt{2 \cdot (2\ell)!} \cdot \left(\frac{8L^2}{km}\right)^{\ell/2} \le \mathbb{P}\{D_{i,i} \ne 0\} \cdot 2^\ell \ell! \cdot \left(\frac{8L^2}{km}\right)^{\ell/2}$$

$$\le \mathbb{P}\{D_{i,i} \ne 0\} \cdot \ell! \cdot \left(\frac{32L^2}{km}\right)^{\ell/2}$$

A union bound and an application of [Theorem C.2](#) (with $\nu = 4L^2 \mathbb{P}\{D_{i,i} \ne 0\}/m$ and $c = 6L/\sqrt{km}$) gives:

$$\left\|\sum_{i=1}^k Z_i\right\| \le 2 \max_{j \in \{1,2\}} \left|\sum_{i=1}^k \langle v_j, Z_i\rangle\right| \le 256 \cdot L \cdot \left(\sqrt{\frac{1}{m} \cdot \mathbb{P}\{D_{i,i} \ne 0\} \cdot \log 1/\delta'} + \frac{\log 1/\delta'}{\sqrt{km}}\right)$$

with probability at least $1 - \delta'$.

For the other term in Eq. (8), we use $U$ to denote the left singular subspace of $M$ and $Y$ to denote the span of the left singular vectors of $\mathcal{P}_V^\perp \mathcal{P}_U$ with non-zero singular values. Noting $\|\mathcal{P}_V^\perp \mathcal{P}_U\| \leq 1$, we now have:

$$\|\widetilde{H}^\top DW\mathcal{P}_V^\perp M\| \leq \|M\| \cdot \|\widetilde{H}^\top DW\mathcal{P}_V^\perp \mathcal{P}_U\| \leq \|M\| \cdot \|\widetilde{H}^\top DW\mathcal{P}_Y\|.$$

We expand the term on the right as follows:

$$\left(\widetilde{H}^\top DW\mathcal{P}_Y\right)^\top = \sum_{i=1}^k \widetilde{H}_i D_{i,i} \mathcal{P}_Y w_i = \sum_{i=1}^k D_{i,i} \cdot \left(\widetilde{H}_i \mathcal{P}_Y w_i\right) =: \sum_{i=1}^k Y_i.$$

Note that $Y$ is orthogonal to $V$ as $\mathcal{P}_V(\mathcal{P}_V^\perp \mathcal{P}_U u) = 0$ for all $u$ and hence, $\widetilde{H}_i \mathcal{P}_Y w_i$ and $D_{i,i}$ are independent random variables (due to $\mathcal{P}_V w_i$ and $\mathcal{P}_V^\perp w_i$ being independent). Now, fix $y \in Y$ s.t $\|y\| = 1$ and we bound the directional even moments of $Y_i$ with the aim of applying Bernstein's inequality as before:

$$\mathbb{E}\left[\langle y, Y_i\rangle^2\right] = \mathbb{E}\left[D_{i,i}^2 \cdot \widetilde{H}_i^2 \cdot \langle y, w_i\rangle^2\right] = \frac{2L^2}{k} \cdot \frac{1}{m} \cdot \mathbb{P}\{D_{i,i} \neq 0\}$$

$$\mathbb{E}\left[\langle y, Y_i\rangle^\ell\right] = \mathbb{E}\left[D_{i,i}^\ell \cdot \widetilde{H}_i^\ell \cdot \langle y, w_i\rangle^\ell\right]$$

$$= ((\ell-1)!!)^2 \cdot \left(\frac{2L^2}{km}\right)^{\ell/2} \cdot \mathbb{P}\{D_{i,i} \neq 0\} \leq \ell! \cdot \left(\frac{2L^2}{km}\right)^{\ell/2} \cdot \mathbb{P}\{D_{i,i} \neq 0\}.$$

Similarly, for odd $\ell \geq 3$, we have by similar manipulations:

$$\mathbb{E}\left[|\langle y, Y_i\rangle|^\ell\right] = \mathbb{E}\left[\left|\widetilde{H}_i^\ell \cdot D_{i,i}^\ell \cdot \langle y, w_i\rangle^\ell\right|\right] \leq \mathbb{P}\{D_{i,i} \neq 0\} \cdot \mathbb{E}\left[|\widetilde{H}_i^\ell \cdot \langle y, w_i\rangle^\ell|\right]$$

$$\leq \mathbb{P}\{D_{i,i} \neq 0\} \cdot \sqrt{\mathbb{E}\left[|\widetilde{H}_i^\ell \cdot \langle y, w_i\rangle^{2\ell}|\right]} \leq \mathbb{P}\{D_{i,i} \neq 0\} \cdot \sqrt{(2\ell)! \cdot \left(\frac{2L^2}{km}\right)^\ell}$$

$$\leq \mathbb{P}\{D_{i,i} \neq 0\} \cdot \sqrt{(2\ell)!} \cdot \left(\frac{2L^2}{km}\right)^{\ell/2} \leq \mathbb{P}\{D_{i,i} \neq 0\} \cdot 2^\ell \ell! \cdot \left(\frac{2L^2}{km}\right)^{\ell/2}$$

$$\leq \mathbb{P}\{D_{i,i} \neq 0\} \cdot \ell! \cdot \left(\frac{8L^2}{km}\right)^{\ell/2}$$

Now, consider a $1/3$-net of $\mathcal{G} := Y \cap \mathbb{S}^{m-1}$. We get for any $z \in Y$:

$$\|z\| = \max_{y \in \mathbb{S}^{m-1}} \langle y, z\rangle = \max_{y \in \mathbb{S}^{m-1}} (\langle y - \widetilde{y}, z\rangle + \langle \widetilde{y}, z\rangle) \leq \frac{\|z\|}{3} + \max_{y \in \mathcal{G}} \langle y, z\rangle \implies \|z\| \leq \frac{3}{2} \max_{y \in \mathcal{G}} \langle y, z\rangle,$$

where $\widetilde{y} = \arg\min_{z \in \mathcal{G}} \|y - z\|$. Since the rank of $Y$ is at most $n$, We may assume $|\mathcal{G}| \leq (30)^n$[Ver18, Corollary 4.2.13]. By a union bound and Theorem C.2:

$$\left\|\sum_{i=1}^n Y_i\right\| \leq 2\max_{y \in \mathcal{G}}\left\langle y, \sum_{i=1}^n Y_i\right\rangle \leq 256L\left(\sqrt{\mathbb{P}\{D_{i,i} \neq 0\} \cdot \left(\frac{n + \log 1/\delta'}{m}\right)} + \left(\frac{n + \log 1/\delta'}{\sqrt{km}}\right)\right)$$

with probability at least $1 - \delta'$. The lemma follows from the previous discussion and Lemma C.5 by picking $\delta' = \delta^\dagger/4$ and applying a union bound. $\qquad\square$

# B  Proof of Theorem 3.1

We will now construct an architecture that when randomly initialized approximately maps every input to a random constant times its Euclidean norm. We will adopt the notation from previous sections; the output of our neural network denoted by $f$ with $\ell$ hidden layers all of fixed width $k$ is defined as follows:

$$f(x) = W_{\ell+1} \cdot \sigma(W_\ell \cdot \sigma(\cdots \sigma(W_1 \cdot x))) \text{ where } \sigma(x)_i = \max\{x_i, 0\}$$

$$f_i(x) = \sigma(W_i \cdot \sigma(\cdots \sigma(W_1 \cdot x)))$$

$$\forall i \in [\ell] \setminus \{1\} : W_i \in \mathbb{R}^{k \times k} \text{ with } (W_i)_{m,n} \overset{iid}{\sim} \mathcal{N}\left(0, \frac{2}{k}\right)$$

$$W_1 \in \mathbb{R}^{k \times d} \text{ with } (W_1)_{m,n} \overset{iid}{\sim} \mathcal{N}\left(0, \frac{2}{d}\right) \qquad W_{\ell+1} \sim \mathcal{N}\left(0, 2I/k\right) \qquad \text{(Lower-Bound-Init)}$$

The scaling for the intermediate layers is chosen such that it preserves the length of the input from the previous layer. We now prove the main result of the section which implies Theorem 3.1 by a simple rescaling:

**Theorem B.1.** *Fix a sufficiently large $d \in \mathbb{N}$, an $\ell \geq d^3$ and $(\ell d)^{20} \leq k \leq \exp(\sqrt{\ell})$, and consider the randomly initialized neural network (Lower-Bound-Init). There is a universal constant $C$ such that with probability at least $0.9$,*

$$\forall x \in \mathbb{S}^{d-1} : |f(x)| \geq 0.04$$

$$\forall x, y \in \mathbb{S}^{d-1} : |f(x) - f(y)| \leq C\sqrt{\frac{\log d}{d}}$$

*Proof.* We start by picking an $\varepsilon$-net of $\mathbb{S}^{d-1}$, $\mathcal{G}$ with $\varepsilon = \left(\frac{1}{10 \cdot 2^\ell}\right)^{10}$. Note we may assume that $|\mathcal{G}| \leq \left(\frac{10}{\varepsilon}\right)^d$. Now, for fixed $x \in \mathcal{G}$ and defining $f_0(x) := x$, we have:

$$\forall i \in [\ell] : \left| \|\sigma(W_i f_{i-1}(x))\| - \|\sigma(W_i' f_{i-1}(x))\| \right| \leq \|\sigma(W_i f_{i-1}(x)) - \sigma(W_i' f_{i-1}(x))\|$$
$$\leq \|W_i - W_i'\| \cdot \|f_{i-1}(x)\| \leq \|W_i - W_i'\|_F \cdot \|f_{i-1}(x)\|.$$

Hence, $\|\sigma(W_i f_{i-1}(x))\|$ is a $\|f_{i-1}(x)\|$-Lipschitz function of $W_i$ and we get by an application of Theorem C.3 (note that $(W_i)_{l,m} \sim \mathcal{N}(0, 2/k)$) and a union bound over the $\ell$ layers:

$$\forall i \in [\ell] : \left| \|f_i(x)\| - \mathbb{E}[\|f_i(x)\| \mid f_{i-1}(x)] \right| \leq 8\|f_{i-1}(x)\| \cdot \sqrt{\frac{\log \delta + \log \ell}{k}}$$

with probability at least $1 - \delta$. By setting $\delta = \frac{1}{16 \cdot |\mathcal{G}| \cdot d^{10}}$ and a union bound over all $x \in \mathcal{G}$, we have with probability at least $1 - 1/(16d^{10})$:

$$\forall x \in \mathcal{G}, i \in [\ell] : \left| \|f_i(x)\| - \mathbb{E}[\|f_i(x)\| \mid f_{i-1}(x)] \right| \leq \|f_{i-1}(x)\| \cdot \frac{1}{2048 \cdot (\ell d)^3}. \tag{9}$$

We also have by Jensen's inequality:

$$\mathbb{E}\left[\|f_i(x)\| \mid f_{i-1}(x)\right] \leq \sqrt{\mathbb{E}\left[\|f_i(x)\|^2 \mid f_{i-1}(x)\right]} = \|f_{i-1}(x)\|$$

and by integrating the tail bound from Theorem C.3:

$$\mathbb{E}\left[\left(\|f_i(x)\| - \mathbb{E}[\|f_i(x)\| \mid f_{i-1}(x)]\right)^2 \mid f_{i-1}(x)\right] \leq \frac{32}{k} \cdot \|f_{i-1}(x)\|^2.$$

Hence, we get:

$$\left(1 - \frac{32}{k}\right) \|f_{i-1}(x)\| \leq \mathbb{E}\left[\|f_i(x)\| \mid f_{i-1}(x)\right] \leq \|f_{i-1}(x)\|.$$

From the above display and Eq. (9) and noting $(1 + x) \leq e^x \leq (1 + 2x)$ for $0 \leq x \leq 1$, we get:

$$\|f_i(x)\| \leq \left(1 + \frac{1}{2048 \cdot (\ell d)^3}\right)^i \leq 1 + \frac{i}{1024\ell^3 d^3}$$

and similarly for the lower bound:

$$\|f_i(x)\| \geq \left(1 - \frac{32}{k} - \frac{1}{2048 \cdot (\ell d)^3}\right)^i \geq 1 - \frac{i}{512\ell^3 d^3}.$$

Putting these together, we obtain:

$$1 - \frac{i}{512\ell^3 d^3} \leq \|f_i(x)\| \leq 1 + \frac{i}{1024\ell^3 d^3} \tag{10}$$

Similarly to the previous discussion, we have for all $x, y \in \mathcal{G}$:

$$\forall i \in [\ell] : |\|\sigma(W_i f_{i-1}(x)) - \sigma(W_i f_{i-1}(y))\| - \|\sigma(W_i' f_{i-1}(x)) - \sigma(W_i' f_{i-1}(y))\||$$
$$\leq \|\sigma(W_i f_{i-1}(x)) - \sigma(W_i f_{i-1}(y)) - \sigma(W_i' f_{i-1}(x)) + \sigma(W_i' f_{i-1}(y))\|$$
$$\leq \|W_i - W_i'\| \cdot (\|f_{i-1}(x)\| + \|f_{i-1}(y)\|) \leq \|W_i - W_i'\|_F \cdot (\|f_{i-1}(x)\| + \|f_{i-1}(y)\|).$$

Therefore, another application of [Theorem C.3](#) and a union bound over the $\ell$ layers yields:

$$\forall i \in [\ell] : |\|f_i(x) - f_i(y)\| - \mathbb{E}[\|f_i(x) - f_i(y)\| \mid f_{i-1}(x), f_{i-1}(y)]|$$
$$\leq 8(\|f_{i-1}(x)\| + \|f_{i-1}(y)\|) \cdot \sqrt{\frac{\log \ell/\delta}{k}}.$$

Setting $\delta = 1/(16 \cdot |\mathcal{G}|^2 \cdot d^{10})$ and a union bound over all $x, y \in \mathcal{G}$ yields with probability at least $1 - 1/(16d^{10})$ for all $x, y \in \mathcal{G}, i \in [\ell]$:

$$|\|f_i(x) - f_i(y)\| - \mathbb{E}[\|f_i(x) - f_i(y)\| \mid f_{i-1}(x), f_{i-1}(y)]| \leq \|f_{i-1}(x) + f_{i-1}(y)\| \frac{1}{2048 \cdot (\ell d)^3}. \tag{11}$$

We only need an upper bound on $\mathbb{E}[\|f_i(x) - f_i(y)\| \mid f_{i-1}(x), f_{i-1}(y)]$. Before, we do we need the following simple fact:

**Fact B.2.** *We have for some $c > 0$:*

$$\forall x \in [0, \pi] : \sin(x) - x \cos(x) \geq \frac{(1 - \cos x)^{3/2}}{15}.$$

*Proof.* Let $f(x) = \sin(x) - x \cos(x)$ we have:

$$\forall x \in \left[0, \frac{\pi}{2}\right] : f'(x) = x \sin(x) \geq \frac{2}{\pi} \cdot x^2$$
$$\forall x \in [0, \pi] : f'(x) = x \sin(x) \geq 0.$$

Therefore, we have:

$$\forall x \in \left[0, \frac{\pi}{2}\right] : f(x) = \int_0^x f'(x)dx \geq \frac{2}{3\pi} \cdot x^3$$
$$\forall x \in \left[\frac{\pi}{2}, \pi\right] : f(x) = \int_0^x f'(x)dx \geq \int_0^{\pi/2} f'(x)dx \geq \frac{\pi^2}{12} \geq \frac{x^3}{36}.$$

By noting that $1 - \cos x \leq x^2/2$, we get:

$$\forall x \in [0, \pi] : f(x) \geq \frac{(1 - \cos(x))^{3/2}}{15}.$$

$\square$

Now, defining $\theta_i = \arccos(\langle f_i(x), f_i(y)\rangle/(\|f_i(x)\|\|f_i(y)\|))$, we have:

$$\mathbb{E}[\|f_i(x) - f_i(y)\| \mid f_{i-1}(x), f_{i-1}(y)]$$
$$\leq \sqrt{\mathbb{E}[\|f_i(x) - f_i(y)\|^2 \mid f_{i-1}(x), f_{i-1}(y)]}$$
$$= \sqrt{\|f_{i-1}(x)\|^2 + \|f_{i-1}(y)\|^2 + 2\mathbb{E}[\langle f_i(x), f_i(y)\rangle \mid f_{i-1}(x), f_{i-1}(y)]}$$
$$= \sqrt{\|f_{i-1}(x)\|^2 + \|f_{i-1}(y)\|^2 - 2\|f_{i-1}(x)\|\|f_{i-1}(y)\| \cdot \left(\frac{\sin \theta_{i-1}}{\pi} + \left(1 - \frac{\theta_{i-1}}{\pi}\right)\cos \theta_{i-1}\right)}$$
$$= \sqrt{\|f_{i-1}(x) - f_{i-1}(y)\|^2 - 2\|f_{i-1}(x)\|\|f_{i-1}(y)\| \cdot \left(\frac{\sin \theta_{i-1} - \theta_{i-1}\cos \theta_{i-1}}{\pi}\right)}$$
$$\leq \sqrt{\|f_{i-1}(x) - f_{i-1}(y)\|^2 - 2\|f_{i-1}(x)\|\|f_{i-1}(y)\| \cdot \left(\frac{\sin \theta_{i-1} - \theta_{i-1}\cos \theta_{i-1}}{\pi}\right)}$$

$$\leq \|f_{i-1}(x) - f_{i-1}(y)\| \cdot \sqrt{1 - \frac{2\|f_{i-1}(x)\|\|f_{i-1}(y)\|}{\|f_{i-1}(x) - f_{i-1}(y)\|^2} \cdot \frac{(1 - \cos\theta_{i-1})^{3/2}}{15\pi}}$$

$$\leq \|f_{i-1}(x) - f_{i-1}(y)\| \cdot \left(1 - \frac{\|f_{i-1}(x)\|\|f_{i-1}(y)\|}{\|f_{i-1}(x) - f_{i-1}(y)\|^2} \cdot \frac{(1 - \cos\theta_{i-1})^{3/2}}{15\pi}\right). \tag{12}$$

On the event in Eq. (10), defining $\widetilde{f}_i(x) = \frac{f_i(x)}{\|f_i(x)\|}$ and similarly for $y$, we have:

$$2(1 - \cos\theta_{i-1})$$
$$= \|\widetilde{f}_{i-1}(x)\|^2 + \|\widetilde{f}_{i-1}(y)\|^2 - 2\langle\widetilde{f}_{i-1}(x), \widetilde{f}_{i-1}(y)\rangle$$
$$\geq \left(1 + \frac{1}{1024\ell^2 d^3}\right)^{-2}(\|f_{i-1}(x)\|^2 + \|f_{i-1}(y)\|^2) - 2\left(1 + \frac{\operatorname{sgn}\cos\theta_{i-1}}{256\ell^2 d^3}\right)\langle f_{i-1}(x), f_{i-1}(y)\rangle$$
$$\geq \|f_{i-1}(x) - f_{i-1}(y)\|^2 - \frac{1}{256\ell^2 d^3}(\|f_{i-1}(x)\| + \|f_{i-1}(x)\|)^2$$
$$\geq \|f_{i-1}(x) - f_{i-1}(y)\|^2 - \frac{1}{48\ell^2 d^3}.$$

Therefore, we get:

$$(1 - \cos\theta_{i-1}) \geq \begin{cases} \frac{1}{4} \cdot \|f_{i-1}(x) - f_{i-1}(y)\|^2, & \text{if } \|f_{i-1}(x) - f_{i-1}(y)\|^2 \geq \frac{1}{16\ell^2 d^3} \\ 0, & \text{otherwise} \end{cases}.$$

By substituting into Eq. (12), we have:

$$\mathbb{E}\left[\|f_i(x) - f_i(y)\| \mid f_{i-1}(x), f_{i-1}(y)\right]$$
$$\leq \|f_{i-1}(x) - f_{i-1}(y)\| \cdot \left(1 - \frac{\|f_{i-1}(x)\|\|f_{i-1}(y)\|}{\|f_{i-1}(x) - f_{i-1}(y)\|^2} \cdot \frac{(1 - \cos\theta_{i-1})^{3/2}}{15\pi}\right)$$
$$\leq \|f_{i-1}(x) - f_{i-1}(y)\| \cdot \begin{cases} \left(1 - \frac{\|f_{i-1}(x) - f_{i-1}(y)\|}{1000}\right), & \text{if } \|f_{i-1}(x) - f_{i-1}(y)\|^2 \geq \frac{1}{16\ell^2 d^3} \\ 1, & \text{otherwise} \end{cases}.$$

By further substituting this into Eq. (11), we have:

$$\forall i \in [\ell] : \|f_i(x) - f_i(y)\| \leq \frac{1}{2048 \cdot (\ell d)^3}$$
$$+ \|f_{i-1}(x) - f_{i-1}(y)\| \cdot \begin{cases} \left(1 - \frac{\|f_{i-1}(x) - f_{i-1}(y)\|}{1000}\right), & \text{if } \|f_{i-1}(x) - f_{i-1}(y)\|^2 \geq \frac{1}{16\ell^2 d^3} \\ 1, & \text{otherwise} \end{cases}.$$

For the rest of the proof, we break into two cases:

**Case 1:** $\|f_j(x) - f_j(y)\|^2 \leq \frac{1}{4\ell^2 d^3}$ for some $j \in [\ell]$. In this case, we simply show that:

$$\forall i \geq [j] : \|f_i(x) - f_i(y)\|^2 \leq \frac{1}{\ell^2 d^3}.$$

Suppose for the sake of contradiction, assume the contrary and let $i^*$ be the least index greater than $j$ such that the above condition was violated. We have:

$$\|f_{i^*}(y) - f_{i^*}(x)\| \leq \frac{1}{2048 \cdot (\ell d)^3}$$
$$+ \|f_{i^*-1}(x) - f_{i^*-1}(y)\| \cdot \begin{cases} \left(1 - \frac{\|f_{i^*-1}(x) - f_{i^*-1}(y)\|}{1000}\right), & \text{if } \|f_{i^*-1}(x) - f_{i^*-1}(y)\|^2 \geq \frac{1}{16\ell^2 d^3} \\ 1, & \text{otherwise} \end{cases}.$$

Now, if $\|f_{i^*-1}(x) - f_{i^*-1}(y)\|^2 \leq \frac{1}{4(\ell^2 d^3)}$, we have:

$$\|f_{i^*}(y) - f_{i^*}(x)\| \leq \frac{1}{2048 \cdot (\ell d)^3} + \frac{1}{2\ell d^{3/2}}$$

yielding the contradiction. Alternatively, we have:

$$\|f_{i^*}(y) - f_{i^*}(x)\| \leq \frac{1}{2048 \cdot (\ell d)^3} + \|f_{i^*-1}(x) - f_{i^*-1}(y)\| - \frac{\|f_{i^*-1}(x) - f_{i^*-1}(y)\|^2}{1000}$$

$$\leq \|f_{i^*-1}(x) - f_{i^*-1}(y)\|$$

yielding a contradiction in this case as well.

**Case 2:** $\|f_j(x) - f_j(y)\|^2 \geq \frac{1}{4\ell^2 d^3}$ for all $j \in [\ell]$. In this case, we have:

$$\forall i \in [\ell] : \|f_i(x) - f_i(y)\| \leq \|f_{i-1}(x) - f_{i-1}(y)\| \cdot \left(1 - \frac{\|f_{i-1}(x) - f_{i-1}(y)\|}{2000}\right).$$

Here, we prove:

$$\|f_\ell(x) - f_\ell(y)\| \leq \frac{1}{\sqrt{\ell d}}.$$

Suppose again for the sake of contradiction that the above condition is violated, then we have:

$$\|f_\ell(x) - f_\ell(y)\| \leq \|x - y\| \cdot \left(1 - \frac{1}{2000\sqrt{\ell d}}\right)^\ell \leq 2 \cdot \exp\left\{-\frac{\sqrt{\ell}}{2000 d}\right\}$$

thus yielding a contradiction.

Therefore, we may assume from [Lemma C.1](#):

$$\forall x, y \in \mathcal{G} : \|f_\ell(x) - f_\ell(y)\| \leq \frac{1}{\sqrt{\ell d}}$$

$$\forall x \in \mathcal{G}, i \in [\ell] : 1 - \frac{i}{512\ell^3 d^3} \leq \|f_i(x)\| \leq 1 + \frac{i}{1024\ell^3 d^3}$$

$$\forall i \in [\ell] : \|W_i\| \leq 4.$$

Furthermore, we have with probability $1 - \delta$:

$$|f(x) - f(y)| = |W_{\ell+1}(f_\ell(x) - f_\ell(y))| \leq 2\|f_\ell(x) - f_\ell(y)\|\sqrt{\log 1/\delta} \leq 2\sqrt{\frac{\log 1/\delta}{\ell d^2}}.$$

By picking $\delta = 1/(64 \cdot d^{10} \cdot |\mathcal{G}|^2)$, we get with probability at least $1 - 1/(64d^{10})$:

$$|f(x) - f(y)| \leq C\sqrt{\frac{\log d}{d}}.$$

For $x \notin \mathcal{G}$, let $\widetilde{x} = \arg\min_{y \in \mathcal{G}} \|x - y\|$ and we have:

$$\|f_\ell(x) - f_\ell(\widetilde{x})\| \leq 4^\ell \varepsilon \leq \frac{1}{2^\ell}$$

and

$$\|f(x) - f(\widetilde{x})\| \leq \frac{1}{2^\ell} \cdot \|W_{\ell+1}\|.$$

On the event, $\|W_{\ell+1}\| \leq 2\sqrt{k}$, this yields the conclusion:

$$\forall x, y \in \mathbb{S}^{d-1} : |f(x) - f(y)| \leq C\sqrt{\frac{\log d}{d}}$$

with probability at least $1 - 1/d^{10}$.

Finally, we have by the anti-concentration of Gaussians that with probability at least $0.95$ for fixed $x \in \mathcal{G}$:

$$|f(x)| \geq 0.05.$$

A union bound and the above two displays concludes the proof. $\square$

## C  Miscellaneous Results

We restate standard results used in our analysis. We start with a fact on Gaussian random matrices.

**Lemma C.1.** *Let $A$ be an $m \times n$ random matrix with $A_{i,j} \overset{i.i.d}{\sim} \mathcal{N}(0,1)$. Then, we have that:*

$$\|A\| \leq 3(\sqrt{m} + \sqrt{n} + \sqrt{\log 1/\delta})$$

*with probability at least $1 - \delta$.*

*Proof.* We may assume without loss of generality that $m \leq n$. Let $\mathcal{G}$ be a $1/3$-grid of $\mathbb{S}^{m-1}$ and we have:

$$\|A\| = \max_{u \in \mathbb{S}^{m-1}} \|u^\top A\| = \max_{u \in \mathbb{S}^{m-1}} \|(u - \widetilde{u})^\top A + \widetilde{u}^\top A\| \leq \frac{\|A\|}{3} + \max_{u \in \mathcal{G}} \|u^\top A\|$$

where $\widetilde{u} = \arg\min_{v \in \mathcal{G}} \|u - \widetilde{u}\|$ which implies:

$$\|A\| \leq \frac{3}{2} \cdot \max_{u \in \mathcal{G}} \|u^\top A\|.$$

Note, we may assume that $|G| \leq (10)^m$ and $u^\top A \sim \mathcal{N}(0, I)$ and we have from [Theorem C.3](#):

$$\forall \|u\| = 1 : \mathbb{P}\left\{\|u^\top A\| \leq \sqrt{n} + \sqrt{\log 1/\delta}\right\} \geq 1 - \delta'.$$

Setting $\delta' = \delta/\mathcal{G}$ and a union bound yields:

$$\max_{u \in \mathcal{G}} \|u^\top A\| \leq \sqrt{n} + \sqrt{m \log(10)} + \sqrt{\log 1/\delta} \leq 2(\sqrt{m} + \sqrt{n} + \sqrt{\log 1/\delta})$$

with probability at least $1 - \delta$ which yields the lemma from the previous discussion. $\square$

We also recall Bernstein's Inequality used frequently throughout our analysis.

**Theorem C.2.** *[BLM13, Theorem 2.1] Let $X_1, \ldots, X_n$ be $n$ independent real-valued random variables. Assume there exist positive numbers $\nu$ and $c$ such that:*

$$\sum_{i=1}^n \mathbb{E}\left[X_i^2\right] \leq \nu \text{ and } \sum_{i=1}^n \mathbb{E}\left[|X_i|^q\right] \leq \frac{q!}{2} \nu c^{q-2} \text{ for all } q \geq 3.$$

*Then, we have:*

$$\mathbb{P}\left\{\sum_{i=1}^n (X_i - \mathbb{E}[X_i]) \geq \sqrt{2\nu t} + ct\right\} \leq e^{-t}.$$

We additionally recall the Tsirelson-Ibragimov-Sudakov inequality.

**Theorem C.3.** *[BLM13, Theorem 5.6] Let $X = (X_1, \ldots, X_n)$ be a vector of $n$ independent standard normal random variables. Let $g : \mathbb{R}^n \to \mathbb{R}$ denote a $L$-Lipschitz function. Then, we have:*

$$\forall t \geq 0 : \mathbb{P}\left\{g(X) - \mathbb{E}g(X) \geq t\right\} \leq \exp\left\{-\frac{t^2}{2L^2}\right\}.$$

We frequently use this inequality with the $1$-Lipschitz functions $g(X) = \pm\|X\|$. Combining Theorem [C.3](#) with Gautschi's inequality, which implies $\mathbb{E}\|X\|/\sqrt{n} \in (\sqrt{1 - 1/n}, \sqrt{1 + 1/n}) \subset (1 - 1/\sqrt{n}, 1 + 1/\sqrt{n})$, gives, for $\delta \in (0, 1)$,

$$\mathbb{P}\left\{\|X\| \geq \sqrt{n} + \sqrt{2 \ln 1/\delta} + 1\right\} \leq \delta, \qquad \mathbb{P}\left\{\|X\| \leq \sqrt{n} - \sqrt{2 \ln 1/\delta} - 1\right\} \leq \delta.$$

This immediately implies the following corollary.

**Corollary C.4.** *For $\delta \in (0, 1)$, a matrix $W_i \in \mathbb{R}^{d_i \times d_{i-1}}$ with independent $\mathcal{N}(0, 1/d_{i-1})$ entries, and vectors $u \in \mathbb{R}^{d_i}$ and $v \in \mathbb{R}^{d_{i-1}}$, both of the following events have probability at least $1 - \delta$:*

$$\left|\|u^\top W_i\| - \|u\|\right| \leq \|u\| \frac{\sqrt{2 \ln(2/\delta)} + 1}{\sqrt{d_{i-1}}}, \quad \left|\|W_i v\| - \|v\|\sqrt{\frac{d_i}{d_{i-1}}}\right| \leq \|v\| \frac{\sqrt{2 \ln(2/\delta)} + 1}{\sqrt{d_{i-1}}}.$$

We prove another simple lemma.

**Lemma C.5.** *Let $x, y \in \mathbb{R}^d, R, r \in \mathbb{R}_+$ be such that $\|x\| \geq R > 0$ and $\|x - y\| \leq r$ with $R \geq r$. Then, we have:*

$$\mathbb{P}_{w \sim \mathcal{N}(0,I)} \left\{ \text{sign}(w^\top x) \neq \text{sgn}(w^\top y) \right\} \leq \frac{3r}{R} \cdot \sqrt{\log R/r}.$$

*Proof.* We have that $X := w^\top x \sim \mathcal{N}(0, \|x\|^2)$ and $Z := w^\top(y - x) \sim \mathcal{N}(0, \|y - x\|^2)$. Hence, we have:

$$\mathbb{P} \left\{ \text{sign}(w^\top x) \neq \text{sign}(w^\top y) \right\} \leq \mathbb{P} \left\{ |Z| \geq |X| \right\}.$$

We have for any $\delta > 0$:

$$\mathbb{P} \left\{ |Z| \geq 2r\sqrt{\log 1/\delta} \right\} \leq \delta \text{ and } \mathbb{P} \left\{ |X| \leq 2r\sqrt{\log 1/\delta} \right\} \leq \frac{4r\sqrt{\log 1/\delta}}{\sqrt{2\pi}R} \leq 2 \cdot \frac{r}{R} \cdot \sqrt{\log 1/\delta}.$$

By a union bound, we have:

$$\forall \delta > 0 : \mathbb{P} \left\{ |Z| \geq |X| \right\} \leq \delta + \frac{2r}{R} \cdot \sqrt{\log 1/\delta}.$$

By setting $\delta = r/R$, we get the conclusion of the lemma. $\qquad \square$

**Lemma C.6.** *Let $d \in \mathbb{N}$ with $d \geq 40$. Then, we have for all $k \in \mathbb{N}, r \geq 0, t \geq 0, M \in \mathbb{R}^{k \times d}$:*

$$\mathbb{P}_{Z \sim \text{Unif}(r\mathbb{S}^{d-1})} \left\{ \|MZ\| \geq t \right\} \leq 2\mathbb{P}_{Y \sim \mathcal{N}(0, 2r^2 I/d)} \left\{ \|MY\| \geq t \right\}.$$

*Proof.* The proof follows from the following manipulations:

$$\begin{aligned}
\mathbb{P}_{Y \sim \mathcal{N}(0, 2r^2 I/d)} \left\{ \|MY\| \geq t \right\} &= \mathbb{E}_{Y \sim \mathcal{N}(0, 2r^2 I/d)} \left[ \mathbf{1} \left\{ \|MY\| \geq t \right\} \right] \\
&\geq \mathbb{E}_{Y \sim \mathcal{N}(0, 2r^2 I/d)} \left[ \mathbf{1} \left\{ \|MY\| \geq t \right\} \mathbf{1} \left\{ \|Y\| \geq r \right\} \right] \\
&= \mathbb{E}_{Y \sim \mathcal{N}(0, 2r^2 I/d)} \left[ \mathbf{1} \left\{ \frac{\|MY\|}{\|Y\|} \geq \frac{t}{\|Y\|} \right\} \mathbf{1} \left\{ \|Y\| \geq r \right\} \right] \\
&\geq \mathbb{E}_{Y \sim \mathcal{N}(0, 2r^2 I/d)} \left[ \mathbf{1} \left\{ \frac{\|MY\|}{\|Y\|} \geq \frac{t}{r} \right\} \mathbf{1} \left\{ \|Y\| \geq r \right\} \right] \\
&\geq \mathbb{E}_{Y \sim \mathcal{N}(0, 2r^2 I/d)} \left[ \mathbb{P}_{Z \sim \text{Unif}(r\mathbb{S}^{d-1})} \left\{ \|MZ\| \geq t \right\} \mathbf{1} \left\{ \|Y\| \geq r \right\} \right] \\
&\geq \mathbb{P}_{Z \sim \text{Unif}(r\mathbb{S}^{d-1})} \left\{ \|MZ\| \geq t \right\} \cdot \mathbb{P}_{Y \sim \mathcal{N}(0, 2r^2 I/d)} \left\{ \|Y\| \geq t \right\} \\
&\geq \frac{1}{2} \cdot \mathbb{P}_{Z \sim \text{Unif}(r\mathbb{S}^{d-1})} \left\{ \|MZ\| \geq t \right\}
\end{aligned}$$

concluding the proof of the lemma. $\qquad \square$

# D Empirical Evaluations

In this section, we provide some experimental evidence for the role played by bottleneck layers in restricting the complexity of intermediate representations and in determining the size of the neighborhood around $x$ where the network retains its near-linear behavior. Here, we restrict ourselves to 1-hidden layer neural networks with the single hidden layer serving as a bottleneck layer. For a 1-hidden layer network with $k$ hidden units parameterized as $f(x) := w_2^\top \text{ReLU}(W_1 x)$ where $w_2 \in \mathbb{R}^k$ and $W_1 \in \mathbb{R}^{k \times d}$, we compute the smallest radius, $r > 0$, such that for any possible value of $\nabla f(y)$ for arbitrary $y \in \mathbb{R}^d$, there exists $w \in \mathbb{R}^d$ with $\|w - x\| \leq r$ and $\nabla f(w) = \nabla f(y)$:

$$r := \inf\{t > 0 : \forall y \in \mathbb{R}^d, \exists w \in \mathbb{R}^d \text{ s.t } \nabla f(w) = \nabla f(y) \text{ and } \|w - x\| \leq t\}$$

A straightforward upper bound on $r$ is the objective value of the following optimization problem:

$$\min_{\delta} \|\delta\|$$

$$\text{s.t } W_1(x - \delta) = 0. \qquad \qquad \text{(Min-Pert)}$$

whose solution is $\|W_1^\dagger W_1 x\|$ where $W_1^\dagger$ denotes the pseudo-inverse of $W_1$. Noting the scale invariance of the ReLU activation function and following Theorem 1.1, we pick $(w_2)_i, (W_1)_{i,j} \stackrel{iid}{\sim} \mathcal{N}(0,1)$. In this setting, $\|W_1^\dagger W_1 x\|$ can be shown to concentrate around $\|x\| \cdot \sqrt{k/d}$ with high probability. Hence, as the width the hidden layer increases, so does complexity of the intermediate representation and consequently, the radius within which the network approximates a linear function around $x$ as the perturbation must "undo" a larger more complex representation. Note that despite the fact that the constraints on $r$ are significantly stronger than the region of near linear behavior guaranteed by Theorem 1.1, this bound is tight up to logarithmic factors. Indeed, Theorem 1.1 guarantees near-linear behavior within a radius of $\|x\| \cdot \widetilde{\Omega}\left(\sqrt{k/d}\right)$ of $x$.

In Fig. 1, we track the optimal length of the perturbation $\delta^*$ relative to the length of the input for a network with input dimension 10000; formally, we plot the value of $\|\delta^*\|/\|x\|$ averaged over 10 runs for the number of hidden units ranging over $\{10, 50, 100, 500, 1000, 5000, 10000\}$. The figure shows that as the number of hidden units are increased, the size of the perturbation required to completely distort the gradient increases. The size of this perturbation is closely related to the complexity of the intermediate representation of the input at the bottleneck layer consistent with Theorem 1.1.

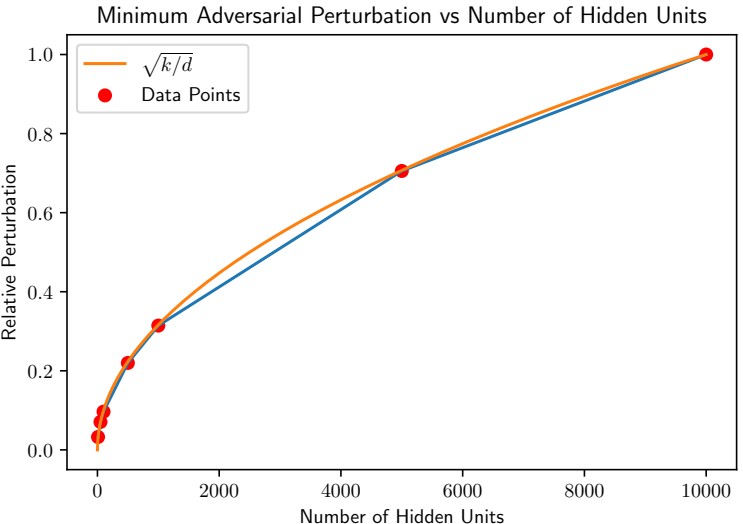

Figure 1: The average value of the optimal solution to Min-Pert as a function of the number of hidden units of a random one-hidden layer neural network with 10000 hidden units averaged over 10 runs. Observe that the data points are in close agreement with the theoretically predicted ratio of $\sqrt{k/d}$.