# OpenReview forum: "Adversarial Examples in Multi-Layer Random ReLU Networks"
_NeurIPS.cc/2021/Conference — NeurIPS 2021 Poster_

### Official Review · Reviewer_2Ydc · 2021-07-12

**Rating:** 6
**Confidence:** 2

**Summary:**

This paper studies how adversarial examples arise in constant depth ReLU networks with independent Gaussian parameters. The main results of the paper is Theorem 1.1 which determines how the width of different network layers affect the probability of the existence of adversarial examples. Overall, the most important takeaway is that smoothness (or the absence of adversarial examples) requires networks that are polynomially (in the number of input dimensions) deeper and/or wider.

**Ethical Concerns:**

No.

**Limitations And Societal Impact:**

I haven't found any discussion about the limitations and/or societal impact. Maybe the authors can highlight some of those in the rebuttal.

**Main Review:**

The paper is well-written and mathematically heavy. While the math seems to work out (I haven't thoroughly gone through every equation), I found no discussion about the proposed theorem.

1) The abstract is unclear. Aside from stating that "adversarial examples arise in these [ReLU] networks [with independent Gaussian parameters] because the functions that they compute are very close to linear", it is difficult to understand what the main theorem implies.

2) The rest of the paper is just as uninformative. I understand what the proof states and that it is a generalization of the results from Daniely and Schacham (2020) and Bubeck et al. (2020), but it is unclear what it implies. A thorough discussion on the implications of the theorem would be welcome (beyond the single paragraph listed after the proof).

3) The related work is missing. Some of it is sprinkled through the paper, but it is again quite difficult to judge the significance of this work without a dedicated section. How does this work relate to empirical work like [MMS+18] or [QMG+19]? How does it relate to over-parametrization and generalization?

4) Finally, for my own understanding, Theorem 3.1. roughtly states that one needs a depth of at least d^3 while Bubeck and Selke (2021) state that "smooth interpolation requires d times more parameters" (where d is the number of inputs dimensions). Can the authors comment on the difference and complementary of the results?

**Time Spent Reviewing:**

3

---

> ### Author Response · Authors · 2021-08-10
> **Initial Author Response**
>
> Thank you for your comments. To address the technical and conceptual contributions of our work, we reproduce the response to reviewer zVxB. The two most closely related to our work are the works of Daniely and Schacham [DS 2020] and Bubeck, Cherapanamjeri, Gidel and Tachet des Combes [BCGdC 2021]. All of these paper establish the presence of adversarial examples in randomly initialized networks with high probability. Daniely and Schacham [DS 2020] establish the result for networks of constant depth with rapidly decreasing width while Bubeck, Cherapanamjeri, Gidel and Tachet des Combes [BCGdC 2021] prove the result for networks with one hidden layer but allowing the width of the hidden layer to be as large as sub-exponential in the dimension of the input. We extend both of these results to the multi-layer setting where we show that the same phenomenon occurs for multi-layer networks of constant depth as long as the width of the widest layer is at most sub-exponential in the width of the narrowest. On a conceptual level, a key contribution of our work is a more nuanced understanding of the role played by bottleneck layers in constraining the complexity of intermediate representations as inputs are propagated through the network which could potentially generalize to the more complicated setting where the network being analyzed is learnt from data. Other related work include the recent result of Bubeck and Sellke [BS 2021] who show that under-parameterized networks trained on random data are not smooth. While these results apply to a broader class of networks including those potentially trained on data, this weaker property on its own does not suffice to explain the prevalence of adversarial examples. Indeed, establishing this fact requires understanding the behavior of the local landscape of the function computed by the network which these approaches do not capture. We see our work as a step towards obtaining a more fine-grained understanding of the local geometry of functions computed by neural networks where we address a natural shortcoming of prior work. The final contribution of our work is in showing that some constraint on depth is necessary for the existence of adversarial examples in this model where we show that sufficiently deep randomly initialized networks do not suffer from adversarial examples.
>
> On a technical level, we establish our result by showing that the gradient at a randomly chosen point is large and that it does not vary significantly in a suitably large radius around the point. Concretely, denoting the function computed by the network as $f$, we show that for some large $r > 0$ and $x \in \mathbb{R}^d$ with $\\|x\\| = \sqrt{d}$:
> \begin{equation*}
>     \forall y\text{ s.t } \\|y - x\\| \leq r: \\|\nabla f(x) - \nabla f(y)\\| = o(1),\ |f(x)| = O(1) \text{ and } \\| \nabla f(x) \\| = \Omega (1).
> \end{equation*}
> Thus, for any such $y$, we additionally have $f(y) - f(x) \approx \langle \nabla f(x), y - x \rangle$. These three properties show that an adversarial example may be obtained from a single gradient descent step of length $O(1)$ from $x$. The key difficulty in our proof is the first inequality establishing the local linearity of $f$. Showing this inequality requires carefully exploiting an architecture specific decomposition of the network between bottleneck layers which play a crucial part in restricting the complexity of intermediate representations as the ball around $x$ is propagated through the network. We believe this insight will have further applications in understanding the behavior of more complicated networks such as those learnt from training data.
>
> We would additionally like to address a misunderstanding regarding the comparison between our work and that of Bubeck and Sellke (2021). The result of Bubeck and Sellke establishes that a (potentially trained) network interpolating random training data must have a point with large gradient. However, as previously remarked, this is not sufficient to establish the existence of adversarial examples. Our result on the other hand does not apply to the setting of learnt networks (and indeed does not feature a randomly generated training set) but establishes a much finer understanding of the local landscape of functions computed by such networks. Therefore, while we show that depth can mitigate the prevalence of adversarial examples in multi-layer networks, the work of Bubeck and Sellke concerns the role of over-parameterization in influencing the robustness of a learnt network. We view these two results as establishing complimentary viewpoints on the same phenomenon and we hope that our results will constitute a step towards a deeper understanding of the phenomenon hinted at by Bubeck and Sellke in learnt networks.
>
> On the relation of our work to the empirical works [MMS+18] and [QMG+19], we note that both these works emphasize the importance of the interplay between the local behavior of the function and the existence of adversarial perturbations. [MMS+18] identify local near-linearity as a cause of adversarial examples and propose a robust training procedure attempting to eliminate this property. On the other hand, [QMG+19] suggest an alternative robust training procedure that retains local near-linearity but eliminates adversarial examples by ensuring that the learnt network (despite being locally linear) is robust to single-step perturbations. The local-linearity of the network suggests that a single step perturbation is close to the optimal adversarial perturbation and hence, such networks do not suffer from adversarial examples. Our work lends theoretical grounding to this phenomenon showing that local linearity arises naturally at initialization of the network. Extending these results to the setting of learnt networks and understanding the effect of robust training procedures on the behavior of the network is a fascinating direction for future research.

---

> > ### Comment · Reviewer_2Ydc · 2021-08-12
> > **Thank you for the clear answer**
> >
> > The authors response (to myself and other reviewers) significantly clarified the paper. I hope the authors can include some of that discussion in the paper revision. In particular, the related work section needs to be more carefully presented and the distinction to prior work (both theoretical and experimental) should be highlighted.
> >
> > I will revise my score accordingly after further discussion with other reviewers.

---

### Official Review · Reviewer_ZKWX · 2021-07-18

**Rating:** 7
**Confidence:** 2

**Summary:**

EDIT: I will be keeping my (positive) score.

The paper studies robustness properties of ReLU networks that have random weights. The main finding is that random ReLU nets of constant depth are susceptible to adversarial examples for every input vector. The main theorem relies on some conditions about the widths in the networks' layers. To complement this result, the authors show that in the large depth regime, such a result cannot hold.

The paper builds on prior work and  their main result generalizes results of Daniely and Schacham (2020) for networks with rapidly decreasing width and results of Bubeck et al (2021) for two-layer networks. Some ideas and intuition come actually from prior works:
the main goal is to demonstrate the existence of an adversarial example near input x,  and to do this for a function f, it turns out that it suffices to show the smoothness property: see (1) . This "smoothness" property is not in the common sense of the word "smooth" function, but basically says that for a deep ReLU network with random parameters and a high-dimensional input vector
x, there is a relatively large ball around x where f satisfies this smoothness property.

The main result is Theorem 1.1 that proves how taking a small step away from a point x in the direction of the gradient can flip the sign of the evaluated function f at x thus corresponding to an adversarial example. The most interesting condition is that the min depth of the net $d_\min \ge c_1(\log d_\max)^{c_2} \log 1/\delta$.

The authors also prove a converse result to justify why their results could only be true in the constant depth regime. This complementary result is given in Theorem 3.1 which shows that when depth grows polynomially in the input dimension d, the function computed by a random ReLU network can essentially be constant, which rules out the possibility of adversarial examples.

For theorem 1, an important aspect of the architecture of the network that allows for adversarial examples or not, depends crucially on the width of the narrowest layer before layer j—that  is "the bottleneck layer" for layer j as the authors call it. This width determines the dimension of the image at layer j of a ball in the input space. In proving bounds on gradients and function values that hold uniformly over pairs of nearby vectors x and y, this dimension—the width of the bottleneck layer—dictates the size of a discretization that is a crucial ingredient in their proof.





**Main Review:**

TL;DR: interesting paper with very nice result on adversarial examples for constant depth ReLU nets

The reviewer finds the paper both technically and conceptually very interesting. It provides a plausible explanation for the abundance of adversarial examples in modern neural nets by giving theory on how such examples would arise. They state a simple property on the gradients that would be sufficient for the existence of attacks and then the authors try to understand how this property would appear in NN. The main result is on constant depth neural nets but the authors complement this by saying the once we allow depth to increase, there can be cases where NN compute functions close to constant hence they don't suffer from adv. examples.

I think the paper is very well-motivated given the lack of understanding for adv. examples and provides a nice set of ideas that will likely be used in future works.


TYPO IN THEOREM 1: sign of f(x) -- I think the authors meant to put sign in front of f

QUESTION to the authors: One might have expected that when the depth is allowed to grow more, then this would lead to even more adversarial examples?  Maybe a naive rationale for this has to do with the lipschitzness of the network that could potentially map x+dx far away from x. My understanding is that because of the randomness in the network, one can avoid adversarial examples as the authors show. But one question that comes to mind is why is it that deep networks in practice tend to suffer from attacks? Is there some difference in the way the initialization or optimization procedures are done?

**Time Spent Reviewing:**

4

---

> ### Author Response · Authors · 2021-08-10
> **Initial Author Response**
>
> Thank you for your thoughtful review of our paper!
>
> We will correct the typo in Theorem 1. Thanks for pointing it out!
>
> Great question! As you point out, it is natural to expect more adversarial examples once the depth of the network increases. However, this fact depends crucially on the scale of the weights chosen on the intermediate layers of the network. If the weights are chosen too small, the output of the network is essentially the $0$ function. On the other hand, if the weights are too large, the output of the network blows up to infinity. Therefore, the choice of normalizing constant is crucial to correctly reason about deep networks. Note that up to normalization, all these networks behave identically due to the positive homogeneity of the $\mathrm{ReLU}$ function. Hence, our analysis shows that if even if the weights of the network are large in which case $x$ and $x + \delta$ are potentially mapped to far away points, they are still close in a relative sense; that is, the relative distance is decreases even when the weights are large, $|f(x + \delta) - f(x)| / |f(x)|$ is small. Unfortunately, it is unclear how these properties evolve through the process of training a network from data. Understanding this phenomenon more deeply is a fascinating direction for future research.

---

### Official Review · Reviewer_BmM8 · 2021-07-24

**Rating:** 7
**Confidence:** 2

**Summary:**

This paper analyzes multi-layer random ReLU networks, and shows theoretical evidence that for constant depth networks with a variety of widths, adversarial examples must necessarily exist. They prove this result based on the key idea that such random networks are nearly linear, and that establishing smoothness accounts for the network’s nonlinearity. Further, they show that adversarial examples will not exist if the depth grows too large.

**Limitations And Societal Impact:**

No. With that said, it is a purely theoretical paper, so I do not believe there is too much for the authors to discuss here.

**Main Review:**

Overall, this paper appears original, has a significantly improved result compared to prior work (extending from 2-layer networks to general constant-depth networks), and is presented in a very clear fashion. The intuition behind each step of the proof is carefully presented in the introductory section, and various proof techniques, such as analyzing the bottleneck layer, are discussed. The proof itself, as well as the key intuitions behind each step of the proof, are also presented in a clear and understandable fashion (there are a lot of important details in the proofs too, which I did not go over carefully). All in all, I would recommend an accept.

I have some suggestions and questions for the authors.
1) Spelling/Grammar - gaussian and euclidean should be capitalized. Heoffding should be Hoeffding.
2) Does the proof still hold when the output dimension is not a single real output? What if it’s a k-dimensional output (e.g. if there are k logits)?
3) This is not necessary, as I believe that the theory presented is an important contribution by itself, but what future experiments (L90) would you suggest to determine if randomly initialized trained networks retain this nearly linear behavior? For example, Lemma 2.2 relies on the gradient being bounded away from 0, but this would probably not be true for a trained neural network, right?


UPDATE: After reading the author response, I will keep my score of Accept, as I still feel that this is an interesting paper.

**Time Spent Reviewing:**

3

---

> ### Author Response · Authors · 2021-08-10
> **Initial Author Response**
>
> Thank you for your careful review of our paper!
>
> Regarding an analogous result for a network with multiple outputs, we believe our proof will extend to the multi-output setting as well. Consider a network with $k$ logit outputs, $f(x): \mathbb{R}^d \to \mathbb{R}^k$, be defined such that $f(x)_i = w_i^\top g(x)$ where $g(x): \mathbb{R}^d \to \mathbb{R}^m$ where $m$ denotes the common embedding of the input used by each logit. In this case, we have $\nabla f(x)_i = w_i^\top \nabla g(x)$. A generalization of our proof allows one to conclude that $\\|\nabla f(x)_i\\| = \Omega(1)$ and that $\langle \nabla f(x)_i, \nabla f(x)_j \rangle \approx 0$ for all $i \neq j$. Furthermore, Lemma 2.8 from our paper applies to each of these components implying that the gradient remains stable over a large radius. Hence, we may find adversarial examples by perturbing the input in the direction of the gradient corresponding to the desired target class.
>
> To establish analogous results for trained networks, we anticipate several challenging difficulties. As you point out, it is not necessary that the gradient is bounded away from $0$ at a fixed input. While recent work by Bubeck and Sellke (2021), does establish results showing that the gradient is large at some point in the input space of the network, this does not establish the existence of adversarial examples. To extend the current results to the trained setting, one needs to establish that the gradient is large at a fixed point and furthermore, it mostly stays large in a suitably large radius around the chosen point. There is empirical evidence for the gradient being large at a random point in previous work by Bubeck, Li and Nagaraj (2021). This suggests the natural empirical experiment of checking whether these randomly chosen points additionally have adversarial examples associated with them.

---

### Official Review · Reviewer_zVxB · 2021-07-27

**Rating:** 5
**Confidence:** 2

**Summary:**

This paper studies some adversarial examples in random ReLu networks, where a smaller perturbation generates a negative label.

The main result of the paper is some theorems. and the main technique used in the paper is mostly some inequality manipulation and concentration. I cannot check all the derivations, but the main results, i.e. the constants are convincible to me.



**Limitations And Societal Impact:**

The experients are ignored in the paper. I get it as common in this field. But still, at least the paper improves upon previous theoretical papers, e.g. it claims that " the minimal width up to some point in the network determines scales and sensitivities of mappings computed up to that point", the authors are encouraged to design an experiment to verify, even without verifying the magnitudes of constants in the theorem.

**Main Review:**

My major concern comes from the applicability of the property it studies. A network with random weights can hardly be associated with any machine learning applications. I can get the point that due to technical difficulty. It is hard to analyze in general, but I think the paper should mention the difference between the random network and a trained network, e.g. if the network is trained, the constants in theorem 1.1. will increase or decrease?



**Time Spent Reviewing:**

1

---

> ### Author Response · Authors · 2021-08-10
> **Initial Author Response**
>
> Thank you for your comments. We would like to start by addressing questions raised about the relation of our work to prior work on this topic. The two most closely related to our work are the works of Daniely and Schacham [DS 2020] and Bubeck, Cherapanamjeri, Gidel and Tachet des Combes [BCGdC 2021]. All of these paper establish the presence of adversarial examples in randomly initialized networks with high probability. Daniely and Schacham [DS 2020] establish the result for networks of constant depth with rapidly decreasing width while Bubeck, Cherapanamjeri, Gidel and Tachet des Combes [BCGdC 2021] prove the result for networks with one hidden layer but allowing the width of the hidden layer to be as large as sub-exponential in the dimension of the input. We extend both of these results to the multi-layer setting where we show that the same phenomenon occurs for multi-layer networks of constant depth as long as the width of the widest layer is at most sub-exponential in the width of the narrowest. On a conceptual level, a key contribution of our work is a more nuanced understanding of the role played by bottleneck layers in constraining the complexity of intermediate representations as inputs are propagated through the network which could potentially generalize to the more complicated setting where the network being analyzed is learnt from data. Other related work include the recent result of Bubeck and Sellke [BS 2021] who show that under-parameterized networks trained on random data are not smooth. While these results apply to a broader class of networks including those potentially trained on data, this weaker property on its own does not suffice to explain the prevalence of adversarial examples. Indeed, establishing this fact requires understanding the behavior of the local landscape of the function computed by the network which these approaches do not capture. We see our work as a step towards obtaining a more fine-grained understanding of the local geometry of functions computed by neural networks where we address a natural shortcoming of prior work. The final contribution of our work is in showing that some constraint on depth is necessary for the existence of adversarial examples in this model where we show that sufficiently deep randomly initialized networks do not suffer from adversarial examples.
>
> On a technical level, we establish our result by showing that the gradient at a randomly chosen point is large and that it does not vary significantly in a suitably large radius around the point. Concretely, denoting the function computed by the network as $f$, we show that for some large $r > 0$ and $x \in \mathbb{R}^d$ with $\\|x\\| = \sqrt{d}$:
> \begin{equation*}
>     \forall y\text{ s.t } \\|y - x\\| \leq r: \\|\nabla f(x) - \nabla f(y)\\| = o(1),\ |f(x)| = O(1) \text{ and } \\| \nabla f(x) \\| = \Omega (1).
> \end{equation*}
> Thus, for any such $y$, we additionally have $f(y) - f(x) \approx \langle \nabla f(x), y - x \rangle$. These three properties show that an adversarial example may be obtained from a single gradient descent step of length $O(1)$ from $x$. The key difficulty in our proof is the first inequality establishing the local linearity of $f$. Showing this inequality requires carefully exploiting an architecture specific decomposition of the network between bottleneck layers which play a crucial part in restricting the complexity of intermediate representations as the ball around $x$ is propagated through the network. We believe this insight will have further applications in understanding the behavior of more complicated networks such as those learnt from training data.
>
> As experimental validation of the role of bottleneck layers, we designed an experiment to show how reducing the width of the hidden layer significantly reduces the complexity of the function computed by the network. Letting the one hidden layer network be denoted by $f(x) = w_2^\top \mathrm{ReLU} (W_1 x)$ where $w_2 \in \mathbb{R}^{k}$ and $W_1 \in \mathbb{R}^{k \times d}$, we compute the length of the smallest perturbation required to obtain any gradient that the network may possibly compute. This length is straightforwardly shown to be the solution to the following optimization problem:
> \begin{equation*}
> \begin{aligned}
>     \min_\delta \\|\delta\\| \\
>     \text{s.t } W (x - \delta) = 0
> \end{aligned}
> \end{equation*}
> whose solution is $W^\dagger W x$ where $W^\dagger$ denotes the pseudo-inverse of $W$. In our experiment, we track the optimal length of the perturbation $\delta$ relative to the length of the input for a one hidden layer neural network with input dimension $10000$; formally, we plot the value of $\\|\delta^*\\| / \\|x\\|$ averaged over $10$ runs of the experiment for the number of hidden units ranging over the set $\\{10, 50, 100, 500, 1000, 5000, 10000\\}$. Our experiment shows that as the number of hidden units are increased, the size of the perturbation required to completely distort the gradient increases. The size of this perturbation is closely related to the complexity of the function computed by the network and intuitively captures the component of the input responsible for the behavior of the function. Hence, the width of the intermediate layer significantly restricts the complexity of the function as suggested by our proof. We will include the results of our experimental evaluation in the paper.

---

### Decision · Program_Chairs · 2021-09-27

**Decision:**

Accept (Poster)

**Comment:**

The paper studies adversarial examples in ReLU networks with independent gaussian parameters. Reviewers were generally happy with the results of the paper. In particular, authors' responses in the rebuttal period helped clarify some concerns. Overall, I think the paper is above the accept threshold.